# The Beijing Climate Center Climate System Model (BCC-CSM): Main Progress from CMIP5 to CMIP6

**Tongwen Wu[1*], Yixiong Lu[1], Yongjie Fang[1], Xiaoge Xin[1], Laurent Li[1,2], Weiping Li[1], Weihua Jie[1], Jie Zhang[1], Yiming Liu[1], Li Zhang[1], Fang Zhang[1], Yanwu Zhang[1], Fanghua Wu[1], Jianglong Li[1], Min Chu[1], Zaizhi Wang[1], Xueli Shi[1], Xiangwen Liu[1], Min Wei[3], Anning Huang[4], Yaocun Zhang[4], Xiaohong Liu[1,5]**

[1]Beijing Climate Center, China Meteorological Administration, Beijing, China

[2]Laboratoire de Météorologie Dynamique, IPSL, CNRS, Sorbonne Université, Ecole Normale Supérieure, Ecole Polytechnique, Paris, France

[3]National Meteorological Information Center, China Meteorological Administration, Beijing, China

[4]Nanjing University, Nanjing, China

[5]University of Wyoming, Laramie, WY, United States

*Correspondence to:* Tongwen Wu (twwu@cma.gov.cn)

**(Revised on Feb. 10, 2019)**

**Abstract**. Main progresses of Beijing Climate Center (BCC) climate system model from the phase five of the Coupled Model Intercomparison Project (CMIP5) to its phase six (CMIP6) are presented, in terms of physical parameterizations and model's performance. BCC-CSM1.1 and BCC-CSM1.1m are the two models involved in CMIP5. BCC-CSM2-MR, BCC-CSM2-HR, and BCC-ESM1.0 are the three models configured for CMIP6. Historical simulations from 1851 to 2014 from BCC-CSM2-MR (CMIP6) and from 1851 to 2005 from BCC-CSM1.1m (CMIP5) are used for models assessment. The evaluation matrices include (a) energy budget at top of the atmosphere, (b) surface air temperature, precipitation, and atmospheric circulation for global and East Asia regions, (c) sea surface temperature

(SST) in the tropical Pacific, (d) sea ice extent and thickness and Atlantic Meridional Overturning
Circulation (AMOC), and (e) climate variations at different time scales such as global warming trend in
the 20$^{th}$ century, stratospheric quasi-biennial oscillation (QBO), Madden-Julian Oscillation (MJO) and
diurnal cycle of precipitation. Compared to BCC-CSM1.1m, BCC-CSM2-MR shows significant
improvements in many aspects including: tropospheric air temperature and circulation at global and
regional scale in East Asia, climate variability at different time scales such as QBO, MJO, diurnal cycle
of precipitation, interannual variations of SST in the equatorial Pacific, and long-term trend of surface
air temperature.

## 1. Introduction

Changes of global climate and environment are main challenges that human societies are facing for
sustainable developments. Climate and environment changes are often the consequence of combined
effects of anthropogenic influences and complex interactions among the atmosphere, hydrosphere,
lithosphere, cryosphere and biosphere of the Earth system. To better understand behaviors of the earth
climate, and to predict its future evolution, appropriate new concepts and relevant methodologies should
be proposed and developed. Climate system models are effective tools to simulate the interactions and
feedbacks in an objective manner, and to explore their impacts on climate and climate change. The
Coupled Model Intercomparison Project (CMIP) organized under the auspices of the World Climate
Research Programme's (WCRP) Working Group on Coupled Modelling (WGCM) started twenty years
ago as a comparison of a handful of early global coupled climate models (Meehl et al., 1997). More
than 30 models participated in the phase five of CMIP (CMIP5, Taylor et al., 2012) and created an
unprecedented dynamics in the scientific community to generate climate information and make them
available for scientific researches. Many of these models were then extended into Earth System models
by including the representation of biogeochemical cycles. BCC effectively contributed to CMIP5 by
running most of the mandatory and optional simulations.
The first generation of Beijing Climate Center ocean-atmosphere Coupled Model BCC-CM1.0 was
developed from 1995 to 2004 (e.g. Ding et al., 2002). It was mainly used for seasonal climate prediction.
Since 2005, BCC initiated the development of a new fully-coupled climate modelling platform (Wu et

al., 2010, 2013, 2014). In 2012, two versions of the BCC model were released: BCC-CSM1.1 with a coarse horizontal resolution T42 (approximately 280 km) and BCC-CSM1.1m with a medium horizontal resolution T106 (approximately 110 km). It was a fully-coupled model with ocean, land surface, atmosphere, and sea-ice components (Wu et al., 2008; Wu, 2012; Xin et al., 2013). Both versions were extensively used for CMIP5. At the end of 2017, the second generation of the BCC model was released to run different simulations proposed by the phase six of CMIP (CMIP6, Eyring et al., 2016). The purpose of this paper is to document the main efforts and progress achieved in BCC for its climate model transition from CMIP5 to CMIP6. We show improvements in both model resolution and its physics. A relevant description on model transition, and experiment design are shown in Sections 2 and 3. A comparison of models performance is presented in Section 4. Conclusions and discussion are summarized in Section 5. Information about code and data availability is shown in Section 6.

## 2. Transition of the BCC climate system model from CMIP5 to CMIP6

Table 1 shows a summary of different BCC models or versions used for CMIP5 and CMIP6. All of them are fully-coupled global climate models with four components, atmosphere, ocean, land surface and sea-ice, interacting with each other. They are physically coupled through fluxes of momentum, energy, water at their interfaces. The coupling was realized with the flux coupler version 5 developed by the National Center for Atmosphere Research (NCAR). BCC-CSM1.1 and BCC-CSM1.1m are our two models involved in CMIP5. They differ mainly by their horizontal resolutions. As shown in Table 1, BCC-CSM2-MR, BCC-CSM2-HR, and BCC-ESM1.0 are the three models developed for CMIP6.

BCC-ESM1.0 is our Earth System configuration. It is a global fully-coupled climate-chemistry-carbon model, and intended to conduct simulations for the Aerosol Chemistry Model Intercomparison Project (AerChemMIP, Collins et al., 2017) and the Coupled Climate–Carbon Cycle Model Intercomparison Project (C4MIP, Jones et al., 2016), both endorsed by CMIP6. Its performance will be presented in a separated paper. BCC-CSM2-HR is our high-resolution configuration prepared for conducting simulations of the High Resolution Model Intercomparison Project (HighResMIP v1.0, Haarsma et al., 2016). It has 56 layers in the vertical, 0.092 hPa for the top of model. Its performance will also be presented separately.

In this paper, we focus on BCC-CSM1.1m and BCC-CSM2-MR. The two models are
representative of our climate modelling efforts in CMIP5 and CMIP6 respectively. They have the same
horizontal resolution (T106, about $110 \times 110$ km in the atmosphere and $30 \times 30$ km in the tropical ocean),
ensuring a fair comparison. But they have different vertical resolutions in the atmosphere (Table 1),
which are 26 layers with its top at 2.917 hPa in BCC-CSM1.1m and 46 layers with its top at 1.459 hPa
in BCC-CSM2-MR (Figure 1). The present version of BCC-CSM2-MR takes 50% more computing
time than BCC-CSM1.1m for the same amount of parallel computing processors.
**2.1 Atmospheric component BCC-AGCM**
The atmospheric component of BCC-CSM1.1m is BCC-AGCM2.2 (second generation). It is
detailed in a series of publications (Wu et al., 2008, 2010; Wu, 2012; Wu et al., 2013).
BCC-AGCM3-MR is its updated version (third generation), used as the atmosphere component in
BCC-CSM2-MR. The dynamic core in the two models is identical and uses the spectral framework
described in Wu et al. (2008), in which a reference stratified atmospheric temperature and a reference
surface pressure are introduced into the governing equations to improve pressure gradient force and
gradients of surface pressure and temperature, the prognostic variables for temperature and surface
pressure are separately replaced by their perturbations from their references. Explicit time difference
scheme is applied to vorticity equation, and semi-implicit time difference scheme for divergence,
temperature, and surface pressure equations. Semi-Lagrangian tracer transport scheme is used for water
vapor, liquid cloud water and ice cloud water. Main differences of model physics used in the two
models (BCC-AGCM2.2 and BCC-AGCM3-MR) are summarized in Table 2 and details in the
following:
*a. Deep convection*
Our second-generation atmospheric model, BCC-AGCM2.2, operates with a parameterization
scheme of deep cumulus convection developed by Wu (2012). Main characteristics can be summarized
as follows:
(1) Deep convection is initiated at the level of maximum moist static energy above the boundary
layer. It is triggered when there is positive convective available potential energy (CAPE) and if the
relative humidity of the air at the lifting level of convective cloud is greater than 75%;
(2) A bulk cloud model taking into account processes of entrainment/detrainment is used to
calculate the convective updraft with consideration of budgets for mass, dry static energy, moisture,
cloud liquid water, and momentum. The scheme also considers the lateral entrainment of the
environmental air into the unstable ascending parcel before it rises to the lifting condensation level. The
entrainment/detrainment amount for the updraft cloud parcel is determined according to the
increase/decrease of updraft parcel mass with altitude. Based on a total energy conservation equation of
the whole adiabatic system involving the updraft cloud parcel and the environment, the mass change for
the adiabatic ascent of the cloud parcel with altitude is derived;
(3) The convective downdraft is assumed to be saturated and originated from the level of minimum
environmental saturated equivalent potential temperature within the updraft cloud;
(4) The closure scheme determining the mass flux at the base of convective cloud is that suggested
by Zhang (2002). It assumes that the increase/decrease of CAPE due to changes of the thermodynamic
states in the free troposphere resulting from convection approximately balances the decrease/increase
resulting from large-scale processes.
A modified version of Wu (2012) is used in BCC-AGCM3-MR for deep convection
parameterization. The convection is triggered only when the boundary layer is unstable or there exists
updraft velocity in the environment at the lifting level of convective cloud, and simultaneously there is
positive CAPE. This modification is aimed to connect the deep convection to the instability of the
boundary layer. The lifting condensation level is set to above the nominal level of non-divergence (600
hPa) in BCC-AGCM2.2 and lowered to the level of 650 hPa in BCC-AGCM3-MR. These modifications
in the deep convection scheme are found to improve the simulation of diurnal cycle of precipitation and
Madden-Julian Oscillation (MJO).
*b. Shallow convection*
Shallow convection is parameterized with a local convective transport scheme (Hack, 1994). It is
used to remove any local instability that may remain after the deep convection scheme. This Hack
convection scheme is largely-used one to typically represent shallow subtropical convection and
midlevel convection that do not originate from the boundary layer.
*c. Cloud macrophysics*
Cloud macrophysics comprises physical processes to compute cloud fractions in each layer,
horizontal and vertical overlapping of clouds, and conversion rates of water vapor into cloud
condensates. In BCC-AGCM2.2, cloud fraction and the associated cloud macrophysics follow what
designed in NCAR Community Atmosphere Model version 3 (CAM3, Collins et al., 2004). The total
cloud cover ($C_{tot}$) within each model grid is set as the maximum value of three cloud covers: low-level
marine stratus ($C_{mst}$), convective cloud ($C_{conv}$), and stratus cloud ($C_s$),
$$C_{tot} = \max\left(C_{conv}, C_{mst}, C_s\right) \tag{1}$$
As in CAM3, the marine stratocumulus cloud is diagnosed with an empirical relationship between the
cloud fraction and the boundary layer stratification which is evaluated with atmospheric variables at
surface and 700mb (Klein and Hartmann, 1993). The convective cloud fraction uses a functional form
of Xu and Krueger (1991) relating the cloud cover to updraft mass flux from the deep and shallow
convection schemes. The stratus cloud fraction is diagnosed on the basis of relative humidity which
varies with pressure.
A new cloud scheme is developed and used in BCC-AGCM3-MR. It consists of calculating
convective cloud and the total cloud cover in a different way from BCC-AGCM2.2. The total cloud
fraction in each model grid cell is given as
$$C_{tot} = C_{conv} + \left(1 - C_{conv}\right)\square\max\left(C_{mst}, C_s\right) \tag{2}$$
And the convective cloud $C_{\text{conv}}$ is assumed to be the sum of shallow ($C_{shallow}$) and deep ($C_{deep}$)
convective cloud fractions:
$$C_{conv} = C_{shallow} + C_{deep} \tag{3}$$
$C_{shallow}$ and $C_{deep}$ are non-overlapped with each other and diagnosed following the relationships,
$$C_{conv}q^*\left(T_c\right) + \left(1 - C_{conv}\right)\overline{q} = \overline{q}_{conv} \tag{4}$$
$$C_{conv}T_c + \left(1 - C_{conv}\right)\overline{T} = \overline{T}_{conv} \tag{5}$$
and
$$q^*\left(T_c\right) = q^*\left(\overline{T}\right) + \frac{\partial q^*\left(\overline{T}\right)}{\partial \overline{T}}\left(T_c - \overline{T}\right) \tag{6}$$
where $\bar{q}$ and $\bar{T}$, $\bar{q}_{conv}$ and $\bar{T}_{conv}$ denote the model grid box-averaged water vapor mixing ratio and
temperature in the 'environment' before and after convection activity, respectively. $T_c$ and $q^*(T_c)$ are
the temperature inside the convective cloud plume and its saturated water vapor mixing ratio. Here, we
assume that the shallow and deep convection can concurrently occur in the same atmospheric column at
any time step. That is, the shallow convection scheme follows the deep convection and occurs at
vertical layers where local instability still remains after deep convection.
If no supersaturation exists in clouds, we can obtain from Eqs. (4) and (5)

$$C_{conv} = \frac{\left(\bar{q}_{conv} - \bar{q}\right) - \dfrac{\partial q^*\left(\bar{T}\right)}{\partial \bar{T}}\left(\bar{T}_{conv} - \bar{T}\right)}{q^*\left(\bar{T}\right) - \bar{q}} . \tag{7}$$

The temperature $T_c$ and the specific humidity $q_c = q^*(T_c)$ of the cloud plume can be firstly derived
from Eqs. (5) and (6). Following the method above, the cloud fraction ($C_{deep}$ and $C_{shallow}$),
temperature ($T_{deep}$ and $T_{shallow}$), specific humidity ($q_{deep}$ and $q_{shallow}$) for the deep convective,
shallow convective clouds can be then deduced sequentially.
After the three moisture processes (i.e. deep convection, then shallow convection, and finally
stratiform precipitation) are finished, the mean temperature ($\bar{T}_{box}$) and specific humidity ($\bar{q}_{box}$) of the
whole model-grid box are then updated. Ambient temperature ($\bar{T}_{ambient}$) and specific humidity
($\bar{q}_{ambient}$) outside convective clouds can be finally estimated using the following Eqs.,

$$\bar{q}_{box} = \bar{q}_{ambient} \cdot \left(1 - C_{deep} - C_{shallow}\right) + q_{deep} \cdot C_{deep} + q_{shallow} \cdot C_{shallow}, \tag{8}$$

and

$$\bar{T}_{box} = \bar{T}_{ambient} \cdot \left(1 - C_{deep} - C_{shallow}\right) + T_{deep} \cdot C_{deep} + T_{shallow} \cdot C_{shallow}. \tag{9}$$

Finally, the stratus cloud fraction $C_S$ is diagnosed on the basis of the relative humidity ($RH_{ambient}$) of
the ambient,

$$C_s = \left(\frac{RH_{ambient} - RH_{min}}{1 - RH_{min}}\right)^2 \tag{10}$$

where $RH_{min}$ is a threshold of relative humidity and $RH_{ambient}$ is derived with $\overline{T}_{ambient}$ and $\bar{q}_{ambient}$.
If $C_{deep} + C_{shallow} > 1$ in Eqs. (8) and (9), $C_{deep}$ and $C_{shallow}$ are scaled to meet the condition
$C_{deep} + C_{shallow} = 1.0$, and then $C_s = 0$. At that condition, we do not calculate $\overline{T}_{ambient}$ and
$\bar{q}_{ambient}$ from Eqs. (8) and (9).
*d. Cloud microphysics*
In BCC-AGCM2.2 and BCC-AGCM3-MR, the essential part of the stratiform cloud microphysics
remains the same and follows the framework of non-convective cloud processes in CAM 3.0 (Collins et
al., 2004) that is the scheme proposed by Rasch and Kristj ánsson (1998) and modified by Zhang et al.
(2003). However there is a noticeable difference of cloud microphysics in the two models concerning
the treatments for indirect effects of aerosols through mechanisms of clouds and precipitation. Indirect
effects of aerosols were not included in BCC-AGCM2.2 for CMIP5. That is, the cloud droplets
effective radius was not related to aerosols, neither the precipitation efficiency. The cloud droplets
effective radius was either prescribed or a simple function of atmospheric temperature. The effective
radius for warm clouds was specified to be 14 μm over open ocean and sea ice, and was a function of
atmospheric temperature over land. For ice clouds, the effective radius was also a function of
temperature following Kristj ánsson et al. (2000).
Aerosol particles influence clouds and the hydrological cycle by their ability to act as cloud
condensation nuclei and ice nuclei. This indirect radiative forcing of aerosols is included in the latest
version of BCC-AGCM3-MR, with the effective radius of liquid water cloud droplets being related to
the cloud droplet number concentration $N_{cdnc}$ ($cm^{-3}$). As proposed by Martin et al. (1994), the
volume-weighted mean cloud droplet radius $r_{l,vol}$ can be expressed as
$$r_{l,vol} = \left[ (3LWC) / \left( 4\pi\rho_w N_{cdnc} \right) \right]^{1/3}, \tag{11}$$

where $\rho_w$ is the liquid water density, LWC is the cloud liquid water content (g $cm^{-3}$). Cloud water
and ice contents are prognostic variables in our model with source and sink terms taking into account
the cloud microphysics. The effective radius of cloud droplets $r_{el}$ is then estimated as
$$r_{el} = \beta \cdot r_{l,vol} \tag{12}$$

where $\beta$ is a parameter dependent on the droplets spectral shape. There are various methods to parameterize it (e.g. Pontikis and Hicks, 1992; Liu and Daum, 2002). We use the calculation proposed by Peng and Lohmann (2003),

$$\beta = 0.00084 N_{cdnc} + 1.22 \tag{13}$$

In BCC-AGCM3-MR, the liquid cloud droplet number concentration $N_{cdnc}$ ($cm^{-3}$) is a diagnostic variable dependent on aerosols mass. It is explicitly calculated with the empirical function suggested by Boucher and Lohmann (1995) and Quaas et al. (2006) :

$$N_{cdnc} = \exp\left[5.1 + 0.41\ln\left(m_{aero}\right)\right] \tag{14}$$

The total aerosols mass is the sum of four types of aerosol,

$$m_{aero} = m_{SS} + m_{OC} + m_{SO_4} + m_{NH_4NO_2} . \tag{15}$$

Here, $m_{aero}$ ($\mu g.m^{-3}$) is the total mass of all hydrophilic aerosols, i.e., the first bin (0.2 to 0.5 $\mu$ m) of sea salt ($m_{SS}$), hydrophilic organic carbon ($m_{OC}$), sulphate ($m_{SO_4}$), and nitrate ($m_{NH_4NO_4}$). Nitrate as a rapidly increasing aerosol species in recent years affects present climate and potentially has large implications on climate change (Xu and Penner, 2012; Li et al., 2014). A dataset of nitrate from NCAR CAM-Chem (Lamarque et al., 2012) is used in our model.

Aerosols also exert impacts on precipitation efficiency (Albrecht, 1989), which is taken into account in the parameterization of non-convective cloud processes. We use the same scheme as in CAM3 (Rasch and Kristj ánsson, 1998; Zhang et al., 2003). There are five processes that convert condensate to precipitate: auto-conversion of liquid water to rain, collection of cloud water by rain, auto-conversion of ice to snow, collection of ice by snow, and collection of liquid by snow. The auto-conversion of cloud liquid water to rain (PWAUT) is dependent on the cloud droplet number concentration and follows a formula that was originally suggested by Chen and Cotton [1987],

$$PWAUT = C_{l,aut} q_l^2 \rho_a / \rho_w \left(\frac{q_l \rho_a}{\rho_w N_{ncdc}}\right)^{1/3} H\left(r_{l,vol} - r_{lc,vol}\right) \tag{16}$$

Where $\hat{q}_l$ is in-cloud liquid water mixing ratio, $\rho_a$ and $\rho_w$ are the local densities of air and water respectively, and

$$C_{l,aut} = 0.55\pi^{1/3} k \left(3/4\right)^{4/3} \left(1.1\right)^4 . \tag{17}$$

In which k = 1.18 $\times$ 10$^6$ cm$^{-1}$ sec$^{-1}$ is the Stokes constant. H(x) is the Heaviside step function with
the definition,
$$H(x) = \begin{cases} 0, & x < 0 \\ 1, & x \geq 0 \end{cases} \tag{18}$$

$r_{lc,vol}$ is the critical value of mean volume radius of the liquid cloud droplets $r_{l,vol}$, and set to 15 µ m.
*e. Gravity wave drag*
Gravity waves can be generated by a variety of sources including orography, convection, and
geostrophic adjustment in regions of baroclinic instability (Richter et al., 2010). Gravity waves
propagate upward from their source regions and break when large amplitudes are attained. This
produces a drag on the mean flow. Gravity wave drag plays an important role in explaining the zonal
mean flow and thermal structure in the upper atmosphere.
In previous versions of BCC models, the orographic gravity wave drag was parameterized as in
McFarlane (1987), but non-orographic sources such as convection and jet-front systems were not
considered. In BCC-AGCM3-MR, the gravity wave drag generated from convective sources is
introduced as in Beres et al. (2004), but drag by frontal gravity waves and orographic blocking effects
are still not involved. The key point of the Beres' scheme is relating the momentum flux phase speed
spectrum to the convective heating properties. In the present version of BCC-AGCM3-MR, the
convective gravity wave parameterization is activated only when the deep convective heating depth is
greater than 2.5 km. Gravity waves generated by topography and fronts are important for higher
latitudes. The efficiency parameter in the McFarlane scheme is set to 0.125 in BCC-AGCM2.2 and
doubled to 0.25 in BCC-AGCM3-MR to obtain a better result of the polar night jet. In future, it is
planned to improve the orographic gravity wave scheme and to implement parameterizations of gravity
waves emitted by fronts and jets.
In the convective gravity wave scheme, the uncertainty in the magnitude of momentum flux arises
from the horizontal scale of the heating and the convective fraction. The convective fraction (CF) within
a grid cell is an important parameter and can be tuned to obtain right wave amplitudes. It is a constant
and valid for all latitudes where convection is active. Previous studies of Alexander et al. (2004) show
that CF can vary from ~0.2% to ~7%–8%. We use 5% in BCC-AGCM3-MR. This parameterization
scheme of convective gravity waves can improve the model's ability to simulate the stratospheric
quasi-biennial oscillation in BCC-AGCM3-MR.
*f. Radiative transfer*
The radiative transfer parameterization in BCC-AGCM2.2 follows the scheme initially
implemented in CAM3 (Collins et al., 2004). Aerosol indirect effects on radiation are not taken into
account and cloud droplets effective radius is only function of temperature for cold clouds and
prescribed to different values for maritime, polar, and continental cases for warm clouds. In
BCC-AGCM3-MR, however, the aerosol indirect effects are fully included and the effective radius of
droplets for liquid clouds is calculated by Equation (12) using the liquid cloud droplet number
concentration.
*g. Boundary layer turbulence*
BCC-AGCM3-MR basically inherits the boundary layer turbulence parameterization used in
BCC-AGCM2.2, which is based on the eddy diffusivity approach (Holtslag and Boville, 1993). The
eddy diffusivity is given by
$$K_c = kw_t z \left(1 - \frac{z}{h}\right)^2, \tag{19}$$

where $w_t$ is a turbulent velocity and $h$ is the boundary layer height, which is estimated as
$$h = z_s + \frac{Ri_c \left\{ \left[u(h) - u_{SL}\right]^2 + \left[v(h) - v_{SL}\right]^2 + \beta u_*^2 \right\}}{(g/\theta_{SL})\left[\theta_v(h) - \theta_{SL}\right]}, \tag{20}$$

where $z_s$ is the height of the lowest model level, $u$, $v$, and $\theta_v$ are horizontal wind components and
virtual potential temperature at height $z$, $u_{SL}$, $v_{SL}$, and $\theta_{SL}$ represent the same variables, but in the
surface layer. $\beta$ in Eq. (20) is a constant and taken as 100. $u_*$ is the friction velocity, and $g$ is
gravitational acceleration.
The critical Richardson number $Ri_c$ in Eq. (20) is a key parameter for calculating the boundary layer
height and is set to a constant (0.3) for all stable conditions in BCC-AGCM2.2. In BCC-AGCM3-MR,
$Ri_c$ varies according to conditions of boundary layer stability to yield more accurate estimates of
boundary layer height, and set to 0.24 for strongly stable conditions, 0.31 for weakly stable conditions,
and 0.39 for unstable conditions based on observational studies of Zhang et al. (2014).
**2.2 Land component BCC-AVIM**
BCC-AVIM, Beijing Climate Center Atmosphere-Vegetation Interaction Model, is a
comprehensive land surface scheme developed and maintained in BCC. The version 1 (BCC-AVIM1.0)
was used as the land component in BCC-CSM1.1m participating in CMIP5 (Wu et al., 2013). It
includes major land surface biophysical and plant physiological processes. Its origin could go back to
the Atmosphere-Vegetation Interaction Model (AVIM) (Ji, 1995; Ji et al., 2008) with the necessary
framework to include biophysical, physiological, and soil carbon-nitrogen dynamical processes. The
biophysical module in BCC-AVIM1.0, with 10 layers for soil and up to five layers for snow, is almost
the same as that used in the NCAR Community Land Model version 3 (CLM3) (Oleson et al., 2004).
The terrestrial carbon cycle in BCC-AVIM1.0 consists of a series of biochemical and physiological
processes modulating photosynthesis and respiration of vegetation. Carbon assimilated by vegetation is
parameterized by a seasonally varying allocation of carbohydrate to leaves, stem, and root tissues as a
function of the prognostic leaf area index. Litter due to turnover and mortality of vegetation, and carbon
dioxide release into atmosphere through the heterogeneous respiration of soil microbes is taken into
account in BCC-AVIM1.0. Vegetation litter falls to the ground surface and into the soil is divided into
eight idealized terrestrial carbon pools according to the timescale of carbon decomposition of each pool
and transfers among different pools, which is similar to that in the carbon exchange between vegetation,
soil and the atmosphere (CEVSA) model (Cao and Woodward, 1998).
BCC-AVIM1.0 has been updated to BCC-AVIM2.0 which serves as the land component of
BCC-CSM2-MR participating in CMIP6. As listed in Table 3, several improvements have been
implemented in BCC-AVIM2.0, such as the inclusion of a variable temperature threshold to determine
soil water freezing/thawing rather than fixed at 0°C, a better calculation of snow surface albedo and
snow cover fraction, a dynamic phenology for deciduous plant function types, and a four-stream
approximation on solar radiation transfer through vegetation canopy. Besides, a simple scheme for
surface fluxes over rice paddy is also implemented in BCC-AVIM2.0. These improvements are briefly
discussed as follows.

(a) Soil water freezes at the constant temperature 0 ℃ in BCC-AVIM1.0, but the actual

freezing-thawing process is a slowly and continuously changing process. We take into account the fact
that the soil water potential remains in equilibrium with the water vapor pressure over pure ice when
soil ice is present. Based on the relationships among soil water matrix potential $\psi$ (mm), soil
temperature and soil water content, a variable temperature threshold for freeze-thaw dependent on soil
liquid water content, soil porosity and saturated soil matrix potential is introduced. The inclusion of this
scheme improves the performance of BCC-AVIM2.0 in the simulation about seasonal frozen soil (Xia et
al., 2011).

(b) In BCC-AVIM1.0, we took into account the snow aging effect on surface albedo with a simple

consideration by using a unified scheme to mimic the snow surface albedo decrease with time. In
BCC-AVIM2.0, we assume different reduction rates of snow albedo with actual elapsed time after
snowfalls in the accumulating and melting stages of a snow season (Chen et al., 2014). Besides, the
variability of sub-grid topography is now taken into account to calculate the snow cover fraction within
a model grid cell.

(c) Unlike the empirical plant leaf unfolding and withering dates prescribed in BCC-AVIM1.0, a

dynamic determination of leaf unfolding, growth, and withering dates according to the budget of
photosynthetic assimilation of carbon similar to the phenology scheme in CTEM (Arora, 2005) was
implemented in BCC-AVIM2.0. Leaf loss due to drought and cold stresses in addition to natural
turnover are also considered.

(d) The four-stream solar radiation transfer scheme within canopy in BCC-AVIM2.0 is based on

the same radiative transfer theory used in atmosphere (Liou, 2004). It adopts the analytic formula of
Henyey-Greenstein for the phase function. The vertical distribution of diffuse light within canopy is
related to transmisivity and reflectivity of leaves, besides, average leaf angle and direction of incident
direct beam radiation influence diffuse light within canopy as well. The upward and downward radiative
fluxes are determined by the phase function of diffuse light, G-function, leaf reflectivity and
transmisivity, leaf area index, and the cosine of solar angle of incident direct beam radiation (Zhou et al.,

2018).

(e) Considering the wide distribution of rice paddies in Southeast Asia and the quite different
characteristics of rice paddies and bare soil, a scheme to parameterize the surface albedo, roughness
length, turbulent sensible and latent heat fluxes over rice paddies is developed (a manuscript is in
preparation) and implemented in BCC-AVIM2.0.
(f) Finally, land-use and land-cover changes are explicitly involved in BCC-AVIM2.0. An increase
in crop area implies the replacement of natural vegetation by crops, which is often known as
deforestation.
**2.3 Ocean and Sea Ice**
There are no significant changes for the ocean and sea ice from BCC-CSM1.1m to
BCC-CSM2-MR. But for the sake of completeness, we present here a short description of them. The
oceanic component is MOM4-L40, an oceanic GCM. It was based on the Z-coordinate Modular Ocean
Model (MOM), version 4 (Griffies, 2005) developed by the Geophysical Fluid Dynamics Laboratory
(GFDL). It has a nominal resolution of $1\,°\times1\,°$ with a tri-pole grid, the actual resolution being from $1/3\,°$
latitude between $30\,°S$ and $30\,°N$ to $1.0\,°$ at $60\,°$ latitude. There are 40 z-levels in the vertical. The two
northern poles of the curvilinear grid are distributed to land areas over Northern America and over the
Eurasian continent. There are 13 vertical levels placed between the surface and the 300-m depth of the
upper ocean. MOM4_L40 adopts some mature parameterization schemes, including Swedy's
tracer-based third order advection scheme, isopycnal tracer mixing and diffusion scheme (Gent and
McWilliams, 1990), Laplace horizontal friction scheme, KPP vertical mixing scheme (Large et al.,
1994), complete convection scheme (Rahmstorf, 1993), overflow scheme of topographic processing of
sea bottom boundary/steep slopes (Campin & Goosse, 1999), and shortwave penetration schemes based
on spatial distribution of chlorophyll concentration (Sweeney et al., 2005).
Concentration and thickness of sea ice are calculated by the Sea Ice Simulator (SIS) developed by
GFDL (Winton, 2000). It is a global sea ice thermodynamic model including the Elastic–Viscous–
Plastic dynamic process and Semtner's thermodynamic process. SIS has 3 vertical layers, including 1
snow cover and 2 ice layers of equal thickness. In each grid, 5 categories of sea ice (including open
water) are considered, according to the thickness of sea ice. It also takes into account the mutual
transformation from one category to another under thermodynamic conditions. The sea ice model
operates on the same oceanic grid and has the same horizontal resolution of MOM_L40. SIS calculates
concentration, thickness, temperature, salinity of sea ice and motions of snow cover and ice sheet. There
is no gas exchange through sea ice.
**2.4. Surface turbulent fluxes between air and sea/sea ice**
The atmosphere and sea/sea ice interplay through the exchange of surface turbulent fluxes of
momentum, heat and water. An optimum treatment of the surface exchange, sound in physics and
economic in computation, is very important in simulating the climate variability. During the past years,
we maintain a continuous effort to improve the turbulent exchange processes between air and sea/sea
ice in different versions of BCC models.
In BCC-CSM1.1m, the bulk formulas of turbulent fluxes over sea surface originate from those
used in CAM3, with some modifications to the roughness lengths and corrections to the temperature
and moisture gradients considering sea spray effects (Wu et al., 2010). The bulk formulas are updated in
BCC-CSM2-MR. The coefficients in roughness lengths calculations were adjusted and the arbitrary
gradient corrections are not used. Instead, a gustiness parameterization is included to account for the
subgrid wind variability that is contributed by boundary layer eddies, convective precipitation, and
cloudiness (Zeng et al., 2002).
In terms of turbulent exchange between air and sea ice, we proposed a new bulk algorithm aiming
to improve flux parameterizations over sea ice (Lu et al., 2013). Based on theoretical and observational
analysis, the new algorithm employs superior stability functions for stable stratification as suggested by
Zeng et al. (1998), and features varying roughness lengths. All the three roughness lengths ($z_0$, $z_T$, $z_Q$) of
sea ice were set to a constant (0.5 mm) in BCC-CSM1.1m. Observational studies show that values of $z_0$
tend to be smaller than 0.5 mm over sea ice in winter and larger than 0.5 mm in summer (Andreas et al.,
2010a; Andreas et al., 2010b). In the new parameterization used in BCC-CSM2-MR, the roughness
lengths for momentum differentiate between warm and cold seasons. For simplicity, $z_0$ is treated as
$$z_0\left(mm\right) = \begin{cases} 0.1 & \text{for } T_s \leq -2°C \\ 0.8 & \text{for } T_s > -2°C \end{cases}, \tag{19}$$

where $T_s$ represents surface temperature. For the scalar roughness lengths, a theoretical-based model
proposed by Andreas (1987) is used in the new scheme. This model expresses the scalar roughness $z_s$
($z_T$ or $z_Q$) as a function of the roughness Reynolds number R*, i.e.,
$$\ln\left(z_s/z_0\right) = b_0 + b_1\left(\ln R_*\right) + b_2\left(\ln R_*\right)^2. \tag{20}$$
Andreas (1987, 2002) tabulates the polynomial coefficients $b_0$, $b_1$ and $b_2$.
**3. Experimental design**
All BCC simulations presented in this work follow the protocols defined by CMIP5 and CMIP6.
We pay attention for them to be comparable in spite of showing the transition of our climate system
model from CMIP5 to CMIP6. The principal simulation to be analyzed is the historical simulation
(hereafter historical) with prescribed forcings from 1850 to 2005 for CMIP5 (to 2014 for CMIP6).
Historical forcings data are based as far as possible on observations and downloaded from the
webpage (https://esgf-node.llnl.gov/search/input4mips/). They mainly include: (1) GHG concentrations
(only $CO_2$, $N_2O$, $Ch_4$, CFC11, CFC12 used in BCC models) with zonal-mean values and updated
monthly; (2) Yearly global gridded land-use forcing; (3) Solar forcing; (4) Stratospheric aerosols (from
volcanoes); (5) CMIP6-recommended anthropogenic aerosol optical properties which is formulated in
terms of nine spatial plumes associated with different major anthropogenic source regions (Stevens et
al., 2017). (6) Time-varying gridded ozone concentrations. In addition, aerosol masses based on CMIP5
(Taylor et al., 2012) are used for on-line calculation of cloud droplet effective radius in BCC model.
The preindustrial initial state of BCC-CSM2-MR is preceded by a 500-years piControl simulation
following the requirement of CMIP6. The initial state of the piControl simulation itself is obtained
through individual spin-up runs of each component of BCC-CSM2-MR in order for piControl
simulation to run stably and fast to its model equilibrium. Actually, the initial states of atmosphere and
land are obtained from a 10-years AMIP run forced with monthly climatology of sea surface
temperature (SST) and sea ice concentration, and the initial states of ocean and sea ice are derived from
a 1000-years forced run with a repeating annual cycle of monthly climatology of atmospheric state from
the Coordinated Ocean-Ice Reference Experiment (CORE) dataset version 2 (Danabasoglu et al., 2014).
Figure 2 shows time series of the annual and global mean of net energy flux at top of the atmosphere
(TOA) and the sea surface temperature for 600 years in the piControl simulation. The whole system in
BCC-CSM2-MR fluctuates around +0.4 W m$^{-2}$ net energy flux at TOA without obvious trend in 600
years. The global mean surface air temperature shows a small warming after 600 years (Fig. 2b). During
the last 300 years, there are ($\pm0.2$ K amplitude) oscillations of centennial scale for the whole globe and
for the areal average of 60 S to 60 N. They are certainly caused by internal variation of the system.

**4.    Evaluation and comparison between BCC CMIP5 and CMIP6 models**

**4.1 Global Energy Budget**

Radiative fluxes at the top of the model atmosphere are fundamental variables characterizing the

Earth's energy balance. Satellite observations in modern time allow us to monitor changes in the net
radiation at top-of-atmosphere (TOA) from 2001 onwards. CERES (Clouds and Earth's Radiant Energy
System) project, with the lessons learned from its predecessor, the Earth's Radiation Budget Experiment
(ERBE), provides improved observation-based data products of Earth's radiation budget (Wielicki et al,
1996). Recently, data of CERES are synthesized with EBAF (Energy Balanced and Filled) data to
derive the CERES-EBAF products, suitable for evaluation of climate models (Loeb et al., 2012). As
shown in Table 4, the TOA shortwave and longwave components in BCC-CSM2-MR are generally
closer to CERES-EBAF compared to those in BCC-CSM1.1m. Model results are for the period 1986–
2005, while the available CERES-EBAF data are for 2003–2014. Globally-averaged TOA net energy is
0.85 W m$^{-2}$ in BCC-CSM2-MR and 0.98 W m$^{-2}$ in BCC-CSM1.1m for the period from 1986 to 2005.
The energy equilibrium of whole earth system in BCC-CSM2-MR is slightly improved.

Clouds constitute a major modulator of the radiative transfer in the atmosphere for both solar and

terrestrial radiations. Their macro and micro properties, including their radiative properties exert strong
impacts on the equilibrium and variation of the radiative budget at TOA or at surface. Figure 3 displays
annual and zonal mean of shortwave, longwave and net cloud radiative forcing for BCC CMIP5 (blue
curves), CMIP6 (red curves) models and observations (black curves). The data used in Fig. 3 are the
same as in Table 4. Although observations and models results cover different time periods, they are still
relevant to reveal climatological mean performance of climate models. In low latitudes between 30 S
and 30 N, BCC-CSM1.1m shows excessive cloud radiative forcing for both shortwave and longwave
radiations. These biases are largely reduced in BCC-CSM2-MR, which is possibly attributed to the new
algorithm of cloud fraction especially for convective cloud amount. Cloud radiative forcing in mid
latitudes shows large uncertainty, also manifested in the large deviation between the two observations.
Cloud radiative forcing in both models is closer to CERES-EBAF than to CERES in mid latitudes. It is
clear that the new physics modifies the simulated climate and cloud properties, including the fractional
coverage of clouds, their vertical distribution as well as their liquid water and ice content.

**4.2. Performance in Simulating the Global Warming in the 20[th] Century**

The historical simulation allows us to evaluate the ability of models to reproduce the global
warming and climate variability in the 20th century. The performance depends on both model
formulation and the time-varying external forcings imposed on the models (Allen et al., 2000). Figure 4
presents global-mean (from 60 ̊S to 60 ̊N) surface air temperature evolutions from HadCRUT4 (Morice
et al., 2012) and BCC CMIP5 and CMIP6 models. Here only the area from 60° S to 60 ̊N is used for
comparison, since few observations existed in polar regions to deduce reliable information in
HadCRUT4, especially before the 20th century. To better reveal long-term trends, the climatological
mean is calculated for the reference period 1961–1990 and removed from the time series. The
interannual variability of both simulations is qualitatively comparable to that observed. When a 11-year
smoothing is applied, the long-term trend of both CMIP6 and CMIP5 models is highly correlated with
HadCRUT4. Figure 4 presents three members of historical simulations from different initial state of the
piControl simulation. The correlation coefficients are 0.90 in CMIP5 and 0.93, 0.93, 0.90 in three
members of CMIP6, respectively.
A remarkable feature in Figure 4 is the presence of a global warming hiatus or pause for the period
from 1998 to 2013 when the observed global surface air temperature warming slowed down. This is a
hot topic, largely debated in the scientific research community (e.g. Fyfe et al., 2016; Medhaug et al.,
2017). Two members (r1i1p1f1 and r2i1p1f1 in Fig. 4) of historical simulations of the CMIP6 model
show a hiatus towards the end of the simulation that resembles the observed one. Although the third
member (r3i1p1f1) simulated a global warming slowdown from 2004 to 2012, it is not comparable to
the observed hiatus as it has a short spell of colder years centered at 2010. Another warming hiatus
occurred in the period of 1942 to 1974. The first and the third members (r1i1p1f1 and r3i1p1f1) of
BCC-CSM2-MR only simulate the warming slowdown in the late period from 1958 to 1974, but the
second member (r2i1p1f1) of BCC-CSM2-MR almost simulate this warming hiatus in the whole period
from 1942 to 1974. So, the simulation of global warming hiatus in BCC CMIP6 model clearly excludes
any simple response to forcing, and makes internal variability a much more likely reason.
The models response of the SAT to volcanic forcing is slightly stronger than that estimated with
HadCRU data. Evident global cooling shocks are coincident with significant volcanic eruptions such as
Krakatoa (in 1883), West Indies Agung (in 1963), and Mount Pinatubo (in 1991). Each of these
volcanic    eruptions    significantly    enriched    stratospheric    aerosols    (available    from
http://data.giss.nasa.gov/modelforce/strataer/). As shown in Figure 4, SAT may decrease by up to 0.4 ℃
within 1 to 2 years after major volcanic eruptions. The substantial cooling response to volcanic
eruptions is, to a great extent, due to the aerosol direct radiative forcing too strong in both versions of
BCC-CSM.
To keep the paper concise and at a reasonable length, only the first member of CMIP6 historical
simulations of BCC-CSM2-MR will be presented hereafter. Biases of annual mean surface air
temperature (at 2 meters) in the whole globe for BCC-CSM2-MR and BCC-CSM1.1m are shown in
Figure 5. In both BCC models, biases are generally within $\pm 3$ ℃, but there are slightly systematic
warm biases over oceans from 50 ℃S to 50 ℃N and systematic cold ones over most land regions in north
of 50 ℃N, in East Asia and in North Africa. Cold biases in high latitudes of the Northern Hemisphere
(North Atlantic, Arctic, North America and Siberia) seem amplified in BCC-CSM2-MR. The land
surface biases in both coupled models are similar to each other. Those patterns of biases are already
present in AMIP simulations (not shown), where effects of oceanic biases are excluded. So those biases
in land surface partly come from their land surface modelling component. In the Southern Ocean, both
models show a strong warm area in the Weddell Sea. BCC-CSM1.1m shows cold biases in other
regions of the Southern Ocean. The disappearance of cold biases in the Southern Oceans in
BCC-CSM2-MR is possibly attributed to the new scheme of cloud fraction (Table 2) as there is a zone
of low-level cloud between 40 ℃S to 60 ℃S in the Southern Ocean (omitted), not only in models but also
in observations.
**4.3 Climate sensitivity to CO$_2$ increasing**
The long trend of global warming in Figure 4 depends on the climate sensitivity which is an
emblematic parameter to characterize the sensitivity of a climate model to external forcing, with all
feedbacks included. It generally designates the variation of global mean surface air temperature in
response to a forcing of doubled CO$_2$ concentration in the atmosphere (IPCC 2013). As commonly
practiced in the climate modelling community, an equilibrium climate sensitivity and a transient climate
response can be separately evaluated, corresponding to a situation of equilibrium and transient states of
climate.

We use the standard simulation of 1% $CO_2$ increase per year (1pctCO$_2$) to calculate the transient

climate response (TCR), while the equilibrium climate sensitivity (ECS) uses the 4xCO$_2$ abrupt-change
simulation by applying the forcing/response regression methodology proposed by Gregory et al. (2004).
The TCR is calculated using the difference of annual surface air temperature between the pre-industrial
experiment and a 20-year period centered on the time of $CO_2$ doubling in 1pctCO$_2$, which is 1.71 for
BCC-CSM2-MR and 2.02 for BCC-CSM1.1m. The ECS is diagnosed from the 150-year run of abrupt
4xCO2 following the approach of Gregory (2012). The method is based on the linear relationship
(Figure 6) governing the changes of net top-of-atmosphere downward radiative flux and the surface air
temperature simulated in abrupt 4xCO$_2$ relative to the pre-industrial experiment. The ECS is equal to a
half of the temperature change when the net downward radiative flux reaches zero (Andrews et al.,
2012). It is assumed here that 2xCO$_2$ forcing is half of that for 4xCO$_2$, hypothesis generally verified in
climate models. As shown in Fig. 6, the ECS is 3.03 for BCC-CSM2-MR and 2.89 for BCC-CSM1.1m.
So the TCR of the new version model BCC-CSM2-MR is lower than BCC-CSM1.1m, but the ECS of
BCC-CSM2-MR is slightly higher than BCC-CSM1.1m.

The linear regression line shown in Figure 6, as pointed out in Gregory et al. (2012), also allows

estimating the instantaneous forcing due to $CO_2$ increase, and eventually feedbacks parameter of the
climate system. The former is the cross point of the linear regression line with Y axis: 6.2 W $m^{-2}$ for
BCC-CSM2-MR and 7.6 W $m^{-2}$ for BCC-CSM1.1m. They can be scaled to the case of 2xCO$_2$ just with
a division factor of 2. Since ECS values are close to each other in the two models, we can easily deduce
that all-feedback factor is larger in BCC-CSM2-MR than in BCC-CSM1.1m. It is actually not
surprising to see differences of 2xCO$_2$ radiative forcing between the two models even the radiative
transfer scheme is kept identical, because changes in 3-D structures of cloud, atmospheric temperature
and water vapor do exert impacts on additional radiative forcing due to $CO_2$ increase in the atmosphere.
It is however interesting to note that feedbacks can operate, in the two models, in such a different way
that ECS keeps almost unchanged between them. We remind that this is a pure coincidence since we did
not intentionally tune our model for its sensitivity.
**4.4 Behaviors of the atmosphere at present day**
The main spatial patterns of observed precipitation climatology are simulated in BCC-CSM1.1m
and BCC-CSM2-MR. Figure 7 shows model biases of annual-mean precipitation for BCC-CSM1.1m
and BCC-CSM2-MR in the globe. They are very close from each other. Their RMSE is also very close:
1.12 mm/day against 1.18 mm/day. Regions of lack of precipitation, such as North India, South China,
the two sides of Sumatra, and the Amazon, experience significant amelioration in the new model.
Excessive rainfalls in Tropical Africa, in the Indian Ocean, in the Maritime Continent seem amplified in
BCC-CSM2-MR. As for the whole globe, the annual mean precipitation coming from convective
process (including deep and shallow convections) accounts for 50% of the total precipitation (2.94
mm/day) in BCC-CSM2-MR and 48% of the total precipitation (2.87 mm/day) in BCC-CSM1.1m. The
convective precipitation increased in BCC-CSM2-MR, and the total amount of precipitation exceeds the
amount (2.68 mm/day) of 1986-2005 mean observed precipitation analyses from Global Precipitation
Climatology Project (Adler et al., 2003). But in some regions such as in the Maritime Continent, stratus
precipitation evidently enhances in BCC-CSM2-MR, where the ratio of convection precipitation to total
precipitation is 39% and even larger than 35% in BCC-CSM1.1m.
We now use the Taylor diagram (Figure 8) to evaluate the general performance of our two models
in terms of temperature at 850hPa, precipitation and atmospheric general circulation. The evaluation is
done against climatology of ERA-Interim dataset for the period of 1986 to 2005 (Dee et al., 2011).
ERA-Interim is the latest global atmospheric reanalysis produced by the European Centre for
Medium‐Range Weather Forecasts (ECMWF).
For global fields, we calculate the spatial pattern correlations between models and ERA-Interim for
the annual-mean climatology of sea level pressure (SLP), temperature at 850 hPa level (T850), zonal
and meridional wind velocity at 850 hPa (U850 and V850), zonal wind velocity at 200 hPa (U200),
geopotential height at 500hPa (Z500), and precipitation from Global Precipitation Climatology Project
(PRCP in Fig. 8, Adler and Chang, 2003) over the period 1980–2000. Except for PRCP and U850
which have lower correlation (less than 0.90) with observation, other variables are all above 0.90 for
their correlation coefficients. The pattern correlation coefficient of Z500 with ERA-Interim is 0.995, the

best correlation among these variables. Except for V850, correlations of all other variables in CMIP6 model version (BCC-CSM2-MR) have an evident improvement compared to CMIP5 version (BCC-CSM1.1m). The normalized standard deviations of most variables except for PRCP and T850 are obviously improved in BCC-CSM2-MR. As a whole, the performances of most variables in BCC-CSM2-MR are better than those in BCC-CSM1.1m.

Results shown in the Taylor diagrams in Figure 8 about improvements in surface climate and atmospheric general circulation at different vertical levels are consistent with improvements in the vertical distribution of atmospheric temperature. Figure 9 shows the yearly-averaged zonal mean of atmospheric temperature biases in BCC-CSM2-MR and BCC-CSM1.1m, with ERA-Interim for the period of 1986–2005 as reference. Overall, both BCC-CSM2-MR and BCC-CSM1.1m have similar biases in their vertical structure, with 1–3 K warmer in the stratosphere (above 100 hPa) for most of the domain equatorward of 70°N and 70°S. There are larger cold biases near the tropopause (centered near 200hPa) for southward of 30°S and northward of 30°N. In the middle to lower troposphere (below 400hPa), there is a warm bias of 1-2K. Improvements in BCC-CSM2-MR are mainly located in the troposphere below 100 hPa. Both cold biases near the tropopause in high latitudes and warm bias in lower latitudes are reduced.

The improvement in tropospheric temperature induces naturally smaller biases for the zonal wind in the whole troposphere in BCC-CSM2-MR (Figure 9). But there are still westerly wind biases of 6 m s$^{-1}$ in the layer of 100-200 hPa in the tropics. Westerly jets at mid-latitudes are slightly too strong in both hemispheres. The zonal mean of zonal wind biases in the high latitudes of the stratosphere in BCC-CSM2-MR have increased near 10 hPa. The too-strong polar night jets clearly indicate an insufficient atmospheric drag at this level. It may be partly caused by the lack of effects in relation to some non-orographic gravity waves generated by atmospheric fronts and jets. We expect to reduce this model bias in next version by adding this process.

Given a much higher vertical resolution and an advanced parameterization of the gravity wave drag, the new model BCC-CSM2-MR is able to represent the stratospheric quasi-biennial oscillation (QBO), as shown in Figure 10 which displays time-height diagrams of the tropical zonal winds averaged from 5°S to 5°N. The three panels show observations from the ERA-Interim reanalysis and

relevant simulation results from the two models in CMIP6 and CMIP5. Figure 10a shows alternative westerlies and easterlies in the lower stratosphere appearing with a mean period of about 28 months in the ERA-Interim reanalysis. In Figure 10b, the BCC-CSM2-MR simulations present a clear quasi-biennial oscillation of the zonal winds as observed. In this study, the QBO period is taken as the time between easterly and westerly wind transitions at 20 hPa. The simulation produces about 12 QBO cycles from 1980 to 2005. The average period is 24.6 months, whereas the shortest and longest cycles last for 18 and 35 months, respectively. ERA-Interim values are 27.9, 23, and 35 months for average, minimum, and maximum of cycle length. The observed asymmetry in amplitude with the easterlies being stronger than the westerlies is reproduced in the simulated zonal winds. At 20 hPa, the simulated easterlies often exceed -20 m s$^{-1}$, while in the reanalysis easterly winds peak at -30 to -40 m s$^{-1}$. Simulated westerlies of the QBO range from 8 to 12 m s$^{-1}$, whereas the reanalysis shows peak winds of 16 to 20 m s$^{-1}$. The amplitudes of the QBO cycles in the simulation are weaker than in the reanalysis, which is possibly due to inadequate gravity wave forcing to drive the QBO. We suspect that the wave-mean flow interaction based on resolved waves such as Kelvin waves and mixed Rossby-gravity waves is probably not performant enough in BCC-CSM2-MR. One reason that would contribute to such a discrepancy is the relatively coarse vertical resolution (Table 1) that would affect the vertical wave lengths and the wave damping process. The downward propagation of the simulated QBO phases occurs in a regular manner, but does not penetrate to sufficiently low altitudes. It may depend on the vertical resolution and the impact of vertical resolution on downward propagation will be discussed in a separate paper. After a few test of model vertical layers, we tend to conclude that 46 vertical layers (Figure 1) seem the minimum number to simulate QBO in BCC-CSM2-MR. In BCC-CSM1.1m, however, as shown in Figure 10c, QBO is inexistent and only a semiannual oscillation of easterlies can be found.

Madden-Julian Oscillation (MJO) is a very important atmospheric variability acting within a periodicity between 20 and 100 days in the tropics with considerable effects on regional weather and climate. It exerts significant impacts on monsoonal circulations and organization of tropical rainfalls. From the tropical Indian Ocean to the Western Pacific, MJO shows a pronounced behavior of eastward propagation, as shown in Figure 11a, in the form of longitude-time, the lagged correlation coefficient of the rainfall in the eastern Indian Ocean (75–85 °E; 5 °S–5 °N) with other positions and with lagged time.

We can easily observe the eastward-propagating characteristic, with a moving velocity estimated at 5 m s$^{-1}$. As shown in a comparison work of Jiang et al. (2015), three fourth of CMIP5 models don't show the propagation behavior, with only a standing oscillation when data are filtered to retain only the 20-100 day variability. Figure 11b and 11c show the same plot, but from our two models in CMIP5 and CMIP6. Although the new model is far from realistic in terms of eastward propagation, there is indeed a clear improvement compared to the old one.

MJO can also exert impacts on weather and climate of extra-tropics, either through emanation of Rossby waves, or the poleward propagation of MJO itself. Figure 11d shows a latitude-time diagram for lagged correlation coefficients when rainfalls are filtered to only retain the variability of 20-100 days. Panels e and f in Figure 11 are the counterpart simulated by our two models. The new model presents a clear improvement.

**4.5 Interannual variation of sea surface temperature (SST) in the equatorial Pacific**

Figure 12 shows the observed and simulated spatial pattern of standard deviation of SST anomalies in the tropical Pacific. Both BCC-CSM2-MR and BCC-CSM1.1m can simulate the position of the strongest variation of SST, situated in the central-eastern Pacific in the east of the dateline. However, cold SST in the eastern equatorial Pacific still extends too far west in both models and a cold tongue bias exists in the equatorial Pacific and even gets a little worse in BCC-CSM2-MR. The annual mean SST in the coldest center near 110°W in the equatorial Pacific is below 23°C in BCC-CSM2-MR, a deterioration compared to BCC-CSM1.1m. As shown in Figure 12a, HadISST observations (Rayner et al., 2003) can clearly identify a zone of large interannual variation of SST from the Peruvian coast to the equatorial cold tongue. It is well simulated in BCC-CSM2-MR, but almost missing in BCC-CSM1.1m.

Figure 13 presents time series of the monthly Nino3.4 SST Index from observations and from simulations of BCC-CSM1.1m and BCC-CSM2-MR. Although amplitudes of interannual variations of the Nino3.4 index in both models are larger than in HadISST observations, it gets weaker in BCC-CSM2-MR with standard deviation of 0.91°C which is close to observation showing standard deviation of 0.79°C. Recent studies of Lu and Ren (2016) reveal that the mean period ENSO in BCC-CSM1.1m is only 2.4 years, much shorter than that in observation. This bias of a too-short

periodicity of the ENSO cycle still persists in BCC-CSM2-MR. Nevertheless, The characteristic of
ENSO irregularity is improved in BCC-CSM2-MR in comparison to BCC-CSM1.1m.

### 4.6 Sea ice state and oceanic overturning circulation

Figure 14 shows time-series of minimum sea-ice extent from 1851 to 2012 for (a) the Arctic in
September and (b) the Antarctic in March as simulated in BCC-CSM2-MR and BCC-CSM1.1m. Base
on Hadley Centre Sea Ice and Sea Surface Temperature data set (Rayner et al., 2003, shown by "Hadley"
in Figure 14), the observed minimum sea-ice extent in each September from 1851 to 2012 gradually
shrinks, especially since the 1960's, as caused by global warming (Figure 4). The extent of Arctic sea
ice in September in BCC-CSM1.1m is about 2x106 km$^2$ smaller than the Hadley Centre data, and it
begins to shrink since the 1910's, earlier than in observations. Although the Arctic sea-ice extent in
September in BCC-CSM2-MR is even further smaller than in BCC-CSM1.1m, the model performance
is improved since the 1960's and becomes closer to the Hadley observation. In Figure 14b, it is to be
noted that the Antarctic minimum sea-ice extent in the new model is very small, almost a half of what
observed. The old model had however a more realistic behavior for this regard. This discrepancy is
related to too-warm temperatures simulated in BCC-CSM2-MR in the Southern Ocean, in particular in
the Weddell Sea. The downward trend in the Arctic summer sea-ice extent is, however, better simulated
in the new model than in the old one.
Figure 15 shows the seasonal cycle of sea ice extent (SIE) and thickness averaged for the period of
1980 to 2005 in the two Polar Regions in our models. Observations of sea ice extent from the Hadley
Centre data and sea ice thickness from the European Centre for Medium-Range Weather Forecasts
(ECMWF) are also plotted for the purpose of comparison. Observations show that the Arctic sea ice
cover reaches a minimum extent of $6.9 \times 10^6$ km$^2$ in September and rises to a maximum extent of
$16.0 \times 10^6$ km$^2$ in March (Fig. 15a). The two models can both capture the seasonal variation and pattern,
but large biases in BCC_CSM1.1m exist in magnitude, especially in boreal winter, which is evidently
improved in BCC-CSM2-MR. As for Antarctic SIE (Fig. 15b), the ice covers in two models also
undergo a very large seasonal cycle, which is similar to observations. However, SIE in BCC-CSM1.1m
is too extensive nearly throughout the year, particularly in southern hemisphere winter. Comparatively,
the new model BCC-CSM2-MR simulates a relatively smaller seasonal cycle than that in BCC-CSM1.1
and closer to observations except in February to March. In terms of ice thickness (Fig. 15c and d), the
two models simulate a thinner ice cover compared to observations in all seasons for both the Arctic and
Antarctic. The most remarkable improvements of BCC-CSM2-MR appear in the Arctic in the boreal
warm seasons, especially from June to September with thicker ice presented in the Arctic Ocean. Those
improvements may be partly achieved with the new model physics such as schemes for turbulent flux
over sea ice and ocean surfaces, cloud fraction, or atmospheric circulation improvements at high
latitudes. But in the Antarctic, the ice thickness in BCC-CSM2-MR gets worse and even much thinner
than that in BCC-CSM1.1m in almost all the year.

The Atlantic Meridional Overturning Circulation (AMOC) plays a significant role in driving the

global climate variation (Caesar et al., 2018). AMOC consists of two primary overturning cells. In the
upper cell, warm water flows northward in the upper 1000 m to supply the formation of the North
Atlantic Deep Water (NADW), which returns southward in the depth range of approximately 1500 to
4000 m. In contrast, in the lower cell, the Antarctic Bottom Water (AABW) flows northward in the
Atlantic basin beneath NADW. Figure 16 shows the time-averaged AMOC simulated by the two
coupled model versions. The two main cells are well depicted. The lower branch of NADW is much
deeper in BCC-CSM2-MR than in BCC-CSM1.1m, as indicated by the depth of the zero-contour line.
Moreover, the central intensity of NADW in BCC-CSM2-MR is over 22.5 Sv about 2.5 Sv stronger
than that in BCC-CSM1.1m, close to observation-based value (25 Sv in Talley et al., 2013).
**4.7 Evaluation of models for their performance in East Asia**

A good simulation of climate over East Asia is always a challenging issue for the climate modelling

community, as the region is under influences   of complex topography (high Tibetan Plateau), and
atmospheric circulations from low latitudes (tropical monsoon circulation) and from higher latitudes.
Figure 17 plots a Taylor diagram to show models performance of main climate variables over East Asia
covering the region (100 °-140 °E, 20 °-50 °N). Both BCC-CSM1.1m (blue figures) and BCC-CSM2-MR
(red figures) are plotted for precipitation, sea-level pressure and variables of the atmospheric general
circulation. There is a clear and remarkable improvement from CMIP5 to CMIP6 in BCC models. The
amelioration is both in the spatial pattern correlation (radial lines) and in the ratio of standard deviations
(circles from the origin).
Figure 18 shows the 1980-2005 climatology of December-January-February and June-July-August
averaged precipitation over China and its surroundings. In boreal winter, GPCP precipitations show a
rain belt from Southeast China to Japan and another rain belt along the southwestern flank of the
Tibetan Plateau. In BCC-CSM1.1m the winter precipitation is too weak in Southeast China and too
strong near Japan, compared to GPCP observations. This rain belt in BCC-CSM2-MR obviously
spreads westward and is much closer to observations. The rain belt along the southwestern flank of the
Tibetan Plateau in BCC-CSM2-MR, however, gets too strong. In boreal summer, large dry biases over
East China are present in BCC-CSM1.1m. Those biases are reduced in BCC-CSM2-MR. The center of
precipitation around Japan is also well simulated in BCC-CSM2-MR.
The East Asian summer monsoon rainfall has a seasonal progression from south to north at the
beginning of summer and then a quick retreat to the south when the summer monsoon terminates (as
shown in Figure 19a). This phenomenon is strongly related to the fact that the East Asian monsoon
rainfall mainly takes place in the frontal zone between warm and humid air mass from the south, and
cold and dry air mass from the north. This seasonal migration is also accompanied with a meridional
movement of the Western North Pacific Subtropical High, an important atmospheric center of action
controlling the climate of the region. In Figure 19 (panels b and c), we compare the two models in terms
of seasonal migration of the monsoon rainfall. In the old model, rainfall was too weak. The new model
produces more precipitation. In terms of seasonal match, both models show a delay of the peak rainfall
by about one month, even longer in BCC-CSM2-MR.
Finally, let us examine the rainfall diurnal cycle in summer. Figure 20 shows the timing of the
rainfall diurnal cycle from observation and the two models. Main zones of nocturne rainfall can be
recognized in the south flank of the Tibetan Plateau, in the Sichuan Basin in the east of the Tibetan
Plateau, and in the north of Xinjiang in Central Asia. There is also a zone of nocturne rainfall in the low
reach of the Yellow River. This is mainly under the influence of nocturne rainfall in the area of the
Bohai Sea. Other regions over land experience diurnal rainfall peak in the afternoon after 16 hours local
time. The diurnal cycle of rainfall was extensively studied in Jin et al. (2013) in terms of physics
causing the diurnal cycle. But the good simulation of diurnal cycle is always a major challenge for
climate modeling. We can see that it is not very well simulated in our old model and in East China the
peak occurs in the mid and later night (0-4 am). But the improvement is quite spectacular in our new
model with rainfall peak delayed in the afternoon. Such an improvement is due to the implementation of
our new trigger scheme in convection parameterization.
**5. Conclusions and discussion**
This paper presents the main progress of BCC climate system models from CMIP5 to CMIP6 and
focuses on the description of CMIP6 version BCC-CSM2-MR and CMIP5 version BCC-CSM1.1m
especially on the model physics. Main updates in model physics include a modification of deep
convection parameterization, a new scheme for cloud fraction, indirect effects of aerosols through
clouds and precipitation, and the gravity wave drag generated by deep convection. Surface processes in
BCC-AVIM have also been significantly improved for soil water freezing treatment, snow aging effect
on surface albedo, and timing of vegetation leaf unfolding, growth, and withering. A four-stream
radiation transfer within the vegetation canopy replaced the two-stream radiation transfer. There is a
new treatment for rice paddy waters. New schemes for surface turbulent fluxes of momentum, heat and
water at the interface of atmosphere and sea/sea ice are also used.
The evaluation of model performance in simulating present-day climatology is presented for main
climate variables, such as, surface air temperature, precipitation, and atmospheric circulation for the
globe and for East Asia. Emphasis is put on comparison between the CMIP5 and CMIP6 model
versions (BCC-CSM2-MR versus BCC-CSM1.1m). The globally-averaged TOA net energy budget is
$0.85 \text{ W} \cdot \text{m}^{-2}$ in BCC-CSM2-MR, and $0.98 \text{ W m}^{-2}$ in BCC-CSM1.1m. Both versions have a very good
energy equilibrium. Model biases of excessive cloud shortwave and longwave radiative forcings over
low latitudes in BCC-CSM1.1m are obviously reduced in BCC-CSM2-MR. When Taylor diagrams are
used to compare the two models for spatial patterns of main climate variables such as 2-meter surface
air temperature, precipitation, and atmospheric general circulation, BCC-CSM2-MR shows an overall
improvement at both global scale and regional scale in East Asia. These improvements in
BCC-CSM2-MR are believed to be achieved by the new scheme of cloud fraction and by the
consideration of indirect effects of aerosol on clouds and precipitation. The cold tongue bias of SST in
the equatorial Pacific in BCC-CSM1.1m still exists in BCC-CSM2-MR. BCC-CSM1.1m has a severe
bias in sea ice extent (SIE) and thickness (Tan et al., 2015): too extensive in cold seasons and less
extensive in warm seasons in both hemispheres. The most impressive improvements in
BCC-CSM2-MR appear in the boreal warm seasons, especially from June to September with thicker ice
presented in the Arctic Ocean. However, in the Southern Hemisphere, the sea ice extent and thickness in
BCC-CSM2-MR become even smaller than those in its previous version. This is still an issue that needs
to be addressed in our future work. There is another model bias of weak oceanic overturning circulation
in BCC-CSM1.1m. This bias is reduced in the new version BCC-CSM2-MR, and the strength of
AMOC is increased.

Further evaluations are performed on climate variabilities at different time scales, including
long-term trend of global warming in the 20[th] century, QBO, MJO, and diurnal cycle of precipitation.
The   globally-averaged annual-mean surface air temperature from the historical simulation of
BCC-CSM2-MR is much closer to HadCRUT4 observation than BCC-CSM1.1m, and the observed
global warming hiatus or warming slowdown in the period from 1998 to 2013 is captured in some
realization of BCC-CSM2-MR. With a higher vertical resolution and inclusion of the gravity wave drag
generated by deep convection, the new version BCC-CSM2-MR is able to reproduce the stratospheric
QBO, while QBO even does not exist in BCC-CSM1.1m. Further investigations on physical
mechanisms controlling QBO simulation in BCC-CSM2-MR will be reported in future. MJO is a very
important atmospheric oscillation at intra-seasonal scales and main features are reproduced and
improved in BCC-CSM2-MR, but with intensity still weaker than its counterpart in observation. At
interannual scale, BCC-CSM1.1m shows too-strong variations of Nino 3.4 SST index, but too-short and
too-regular periodicity for ENSO. BCC-CSM2-MR shows weaker amplitude for Nino 3.4 SST index,
which is an improvement and closer to HadISST observations. The rainfall diurnal cycle in China has
strong regional variations with pronounced nocturne rainfalls in the Sichuan Basin and in north China
near the Bohai Sea and the coast. The diurnal rainfall generally peaks in the local time afternoon for
most other land regions. BCC-CSM2-MR shows a clear improvement of rainfall diurnal peaks
compared to the CMIP5 model (BCC-CSM1.1m). This improvement of rainfall diurnal variation is
strongly related to the modification of deep convection scheme.

Finally, we also evaluate the climate sensitivity to $CO_2$ increasing in the standard simulation of 1%
$CO_2$ increase per year (1pctCO2) and the 4xCO2 abrupt-change. The transient climate response in the

new CMIP6 model version BCC-CSM2-MR is lower than that in the previous CMIP5 model BCC-CSM1.1m, while the equilibrium climate sensitivity ECS for BCC-CSM2-MR is slightly higher than its counterpart in BCC-CSM1.1m.

From our model evaluations, we find that although basic feature of the QBO can be simulated in BCC-CSM2-MR, the magnitude between westerly and easterly interchange is still too weak. We also note that there are large biases of air temperature and winds in the stratosphere. Therefore, improvement of the stratospheric temperature and circulation simulations is an important priority in the future development of BCC models. In addition, sea ice simulation in the Antarctic region has large biases, which need to be improved.

**6.  Code and data availability**

Source codes of BCC models are freely available upon request addressed to Tongwen Wu (twwu@cma.gov.cn). Model output of BCC models for both CMIP5 and CMIP6 simulations described in this paper is distributed through the Earth System Grid Federation (ESGF) and freely accessible through the ESGF data portals after registration. Details about ESGF are presented on the CMIP Panel website at http://www.wcrp-climate.org/index.php/wgcm-cmip/about-cmip.

**Author contributions**

Tongwen Wu led the BCC-CSM development. Tongwen Wu and Xiaoge Xin designed the experiments and carried them out. Tongwen Wu, Laurent Li, and Xiaohong Liu wrote the final document with contributions from all other authors.

**Acknowledgements**

This work was supported by The National Key Research and Development Program of China (2016YFA0602100). Two anonymous reviewers are acknowledged for their constructive comments on earlier versions of the paper.

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

## Table 1. BCC models for CMIP5 and CMIP6

| Model versions | Atmosphere | Atmos Chemistry and Aerosol | Land Surface | Ocean | Sea Ice |
|---|---|---|---|---|---|
| BCC-CSM1.1 in CMIP5 (Wu et al., 2013) | BCC-AGCM2.1 (1) T42,26 layers (2) Top at 2.917 hPa | (1) Prescribed aerosols (2) No atmospheric chemistry (3) Global carbon budget without spatial distribution | BCC-AVIM1.0 | MOM4-L40v1 (1) Tri-polar: 0.3 to 1 deg latitude x 1 deg longitude, and 40 layers (2) Oceanic carbon cycle based on OCMIP2 | SISv1 |
| BCC-CSM1.1m in CMIP5 (Wu et al., 2013) | BCC-AGCM2.2 (1) T106,26 layers (2) Top at 2.917 hPa | Same as BCC-CSM1.1 | BCC-AVIM1.0 | MOM4-L40v2 | SISv2 |
| BCC-CSM2-MR In CMIP6 | BCC-AGCM3-MR (1) T106,46 layers (2) Top at 1.459 hPa | (1) Prescribed aerosols (2) No atmospheric chemistry (3) Prognostic spatial $CO_2$ in the atmosphere | BCC-AVIM2.0 | MOM4-L40v2 | SISv2 |
| BCC-CSM2-HR In CMIP6 | BCC-AGCM3-HR (1) T266,56 layers (2) Top at 0.092 hPa | (1) Prescribed aerosols (2) No atmospheric chemistry | BCC-AVIM2.0 | MOM4-L40v2 | SISv2 |
| BCC-ESM1 In CMIP6 | BCC-AGCM3-Chem (1) T42,26 layers (2) Top at 2.917 hPa | (1) Prognostic aerosols (2) MOZART2 atmospheric chemistry | BCC-AVIM2.0 | MOM4-L40v2 | SISv2 |

Table 2. Main physics schemes in atmospheric components (BCC-AGCM) of BCC-CSM versions for CMIP5 and CMIP6

| | **BCC-AGCM2 for CMIP5** | **BCC-AGCM3 for CMIP6** |
|---|---|---|
| **Deep convection** | The cumulus convection parameterization scheme (Wu, 2012) | A modified Wu'2012 scheme described in this work |
| **Shallow/Middle Tropospheric Moist Convection** | Hack (1994) | Hack (1994) |
| **Cloud macrophysics** | Cloud fraction diagnosed from updraft mass flux and relative humidity (Collins et al., 2004) | A new scheme to diagnose cloud fraction described in this work |
| **Cloud microphysics** | Modified scheme of Rasch and Kristj ánsson (1998) by Zhang et al. (2003). No aerosol indirect effects | Modified scheme of Rasch and Kristj ánsson (1998) by Zhang et al. (2003), but included the aerosol indirect effects in which liquid cloud droplet number concentration is diagnosed using the aerosols masses. |
| **gravity wave drag** | Gravity wave drag only generated by orography (Mcfarlane 1987) | Gravity wave drag generated by both orography (Mcfarlane 1987) and convection (Beres et al., 2004) using tuned parameters related to model resolutions. |
| **Radiative transfer** | Radiative transfer scheme used in CAM3 (Collins et al., 2004) with no aerosol indirect effects, and cloud drop effective radius for clouds is only function of temperature and has a distinct difference between maritime, polar, and continental for warm clouds. | Radiative transfer scheme used in CAM3 (Collins et al., 2004), but including the aerosol indirect effects, and the effective radius of the cloud drop for liquid clouds is diagnosed using liquid cloud droplet number concentration. |
| **Boundary Layer** | ABL parameterization [Holtslag and Boville, 1993] | ABL parameterization [Holtslag and Boville, 1993], but modified PBL height computation referred to Zhang et al. (2014) |

Table 3. Main physics schemes in BCC-AVIM versions

| BCC-AVIM1.0 in CMIP5 | BCC-AVIM2.0 in CMIP6 |
|---|---|
| ◆ Soil-Vegetation-Atmosphere Transfer module | ◆ Modified freeze-thaw scheme for soil water (below 0 degree and dependent on soil & water) (Xia et al., 2011) |
| ◆ Multi-layer snow-soil scheme (same as NCAR CLM3) | ◆ Improved parameterization of snow surface albedo (Chen et al., 2014) and snow cover fraction (Wu and Wu, 2004) |
| ◆ Snow Cover Fraction scheme (sub-grid topography) | ◆ Four-stream radiation transfer through vegetation canopy (Zhou et al., 2018) |
| ◆ Vegetation growth module | ◆ A vegetation phenology similar to Canadian Terrestrial Ecosystem Model (Arora and Boer, 2005) |
| ◆ Soil carbon decomposition module | |
| ◆ Land use change module (variable crop planting area) | ◆ Parameterized rice paddy scheme |
| | ◆ land VOC module (Guenther et al., 2012) |

Table 4. Energy balance and cloud radiative forcing at the top-of-atmosphere (TOA) in the model with contrast to CERES/EBAF and CERES observations. Units: W $\cdot$ m$^{-2}$.

| | BCC-CSM2-MR (CMIP6) | BCC-CSM1.1m (CMIP5) | CERES/EBAF (OBS) | CERES (OBS) |
|---|---|---|---|---|
| Net energy at TOA | 0.85 | 0.98 | 0.81 | 5.73 |
| TOA outgoing longwave radiative flux | 239.15 | 236.10 | 239.72 | 238.95 |
| TOA incoming shortwave Radiation | 340.46 | 341.70 | 340.18 | 341.47 |
| TOA net shortwave radiative flux | 239.09 | 235.96 | 240.53 | 244.68 |
| TOA outgoing longwave radiative flux in clear sky | 265.02 | 265.58 | 265.80 | 266.87 |
| TOA net shortwave radiative flux in clear sky | 288.67 | 288.71 | 287.68 | 294.69 |
| Shortwave cloud radiative forcing | -49.55 | -52.71 | -47.16 | -48.58 |
| Longwave cloud radiative forcing | 25.87 | 29.48 | 26.07 | 27.19 |

Notes: The model data are the mean of 1986 to 2005, while the available observation data are for 2003–2014.

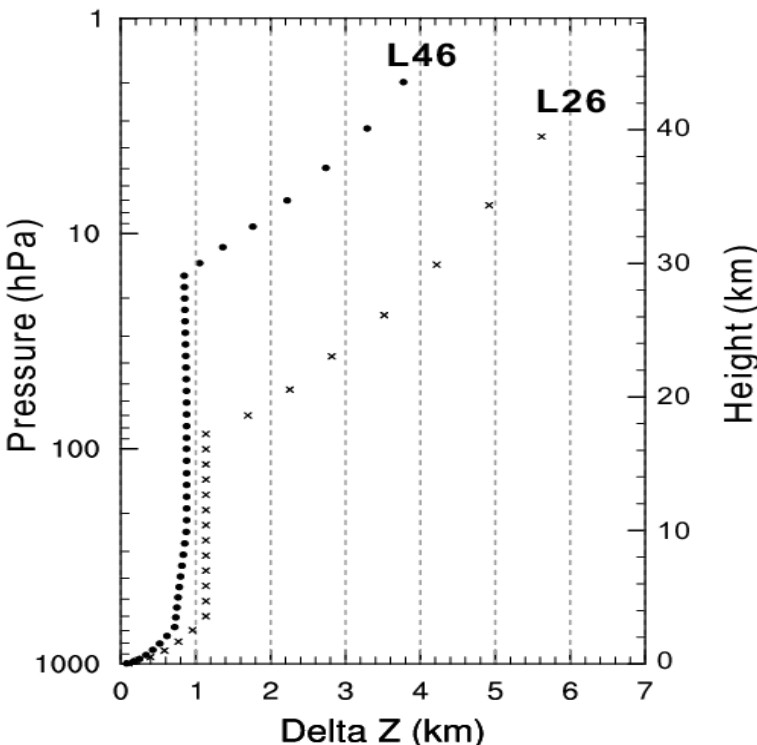

Figure 1. The profiles of layer thickness against the height for 26 vertical layers of the atmosphere in BCC-CSM-1.1m and 46 vertical layers in BCC-CSM2-MR.

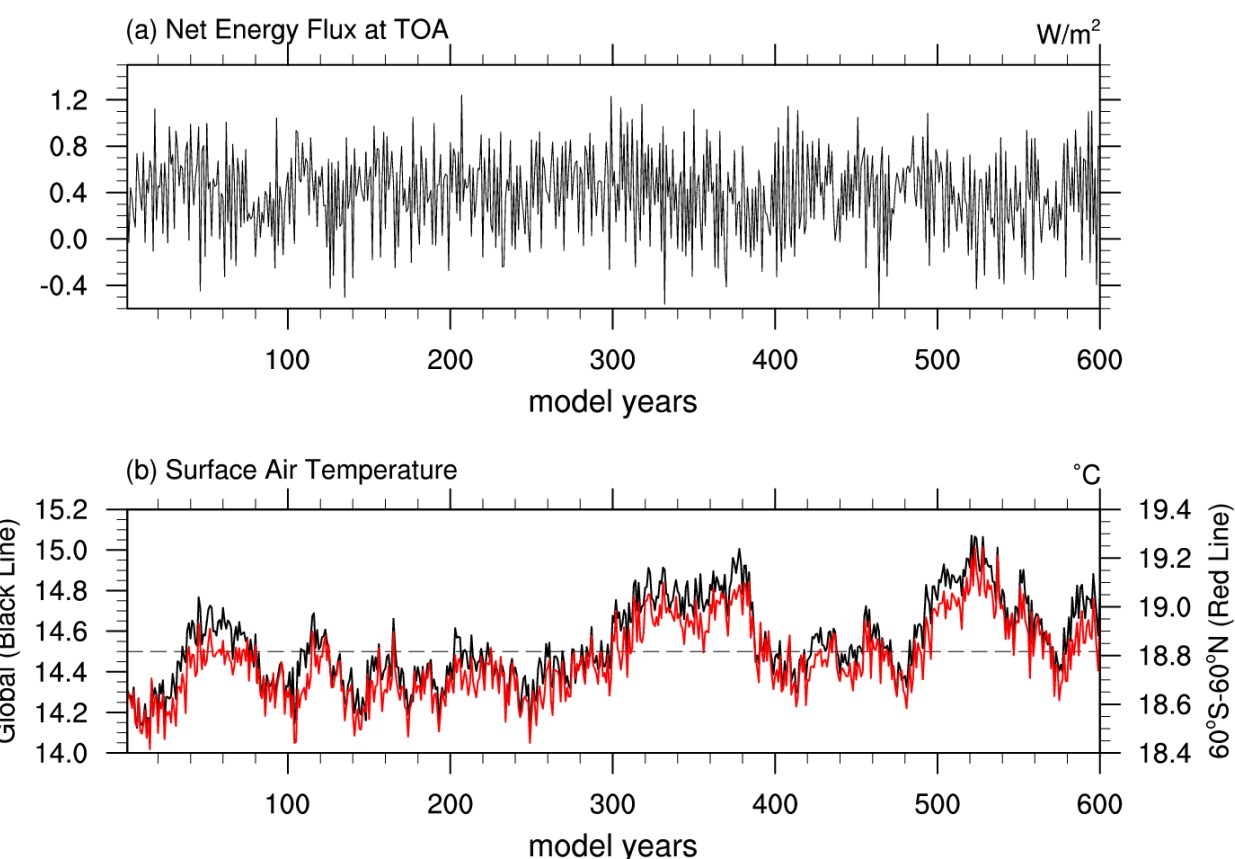

Figure 2. The time series of (a) global mean net energy flux at top of the atmosphere (W m$^{-2}$) and (b) global (black line) and regional (60 °S to 60 °N, red line) surface air temperature (℃) for the 600 years of piControl simulations.

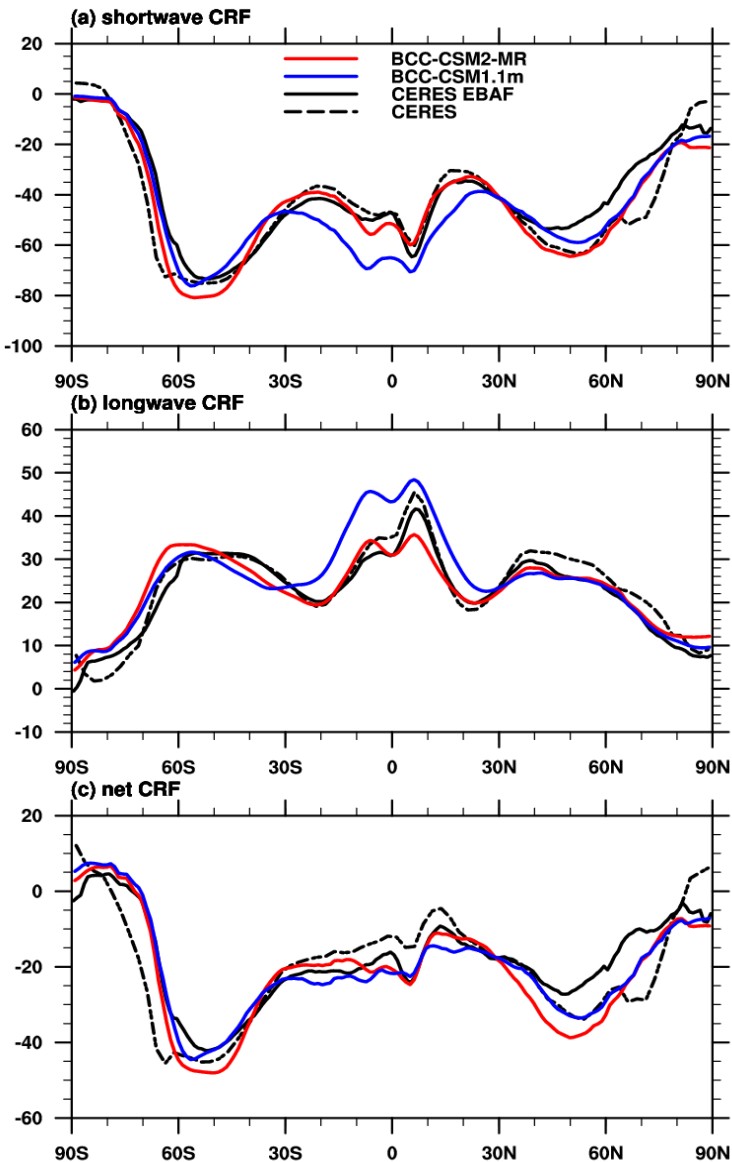

Figure 3. Zonal averages of the cloud radiative forcing from the BCC CMIP5 and CMIP6 models and observations (in W $m^{-2}$; top row: shortwave effect; middle row: longwave effect; bottom row: net effect). Model results are for the period 1986–2005, while the available CERES ES-4 and CERES EBAF 2.6 data set are for 2003–2014.

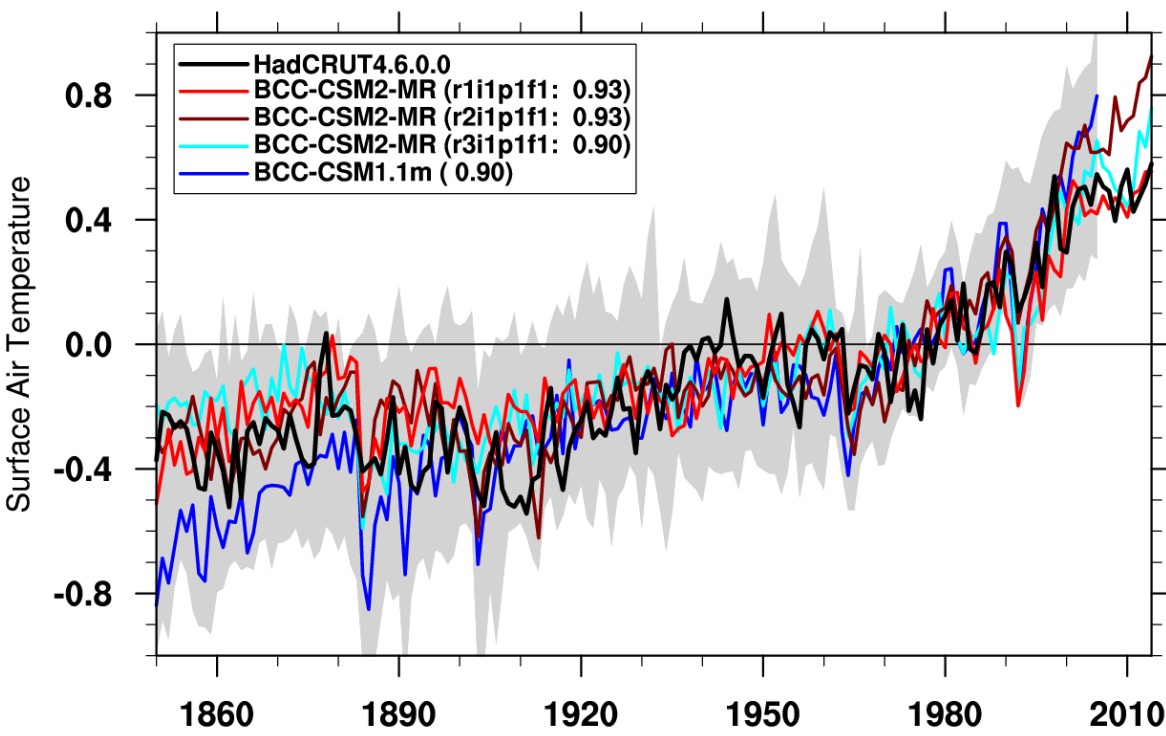

Figure 4. Time series of anomalies in the global (60 °S to 60 °N) mean surface air temperature from 1850 to 2014. The reference climate to deduce anomalies is for each individual curve from 1961 to 1990. Three lines labeled BCC-CSM2-MR denote three members of historical simulations from different initial state of the piControl simulation. The numbers in the bracket denote the correlation coefficient of 11-year smoothed BCC model data with the HadCRUT4.6.0.0 (Morice et al., 2012) observation. Gray shaded area shows the spread of 36 CMIP5 models data.

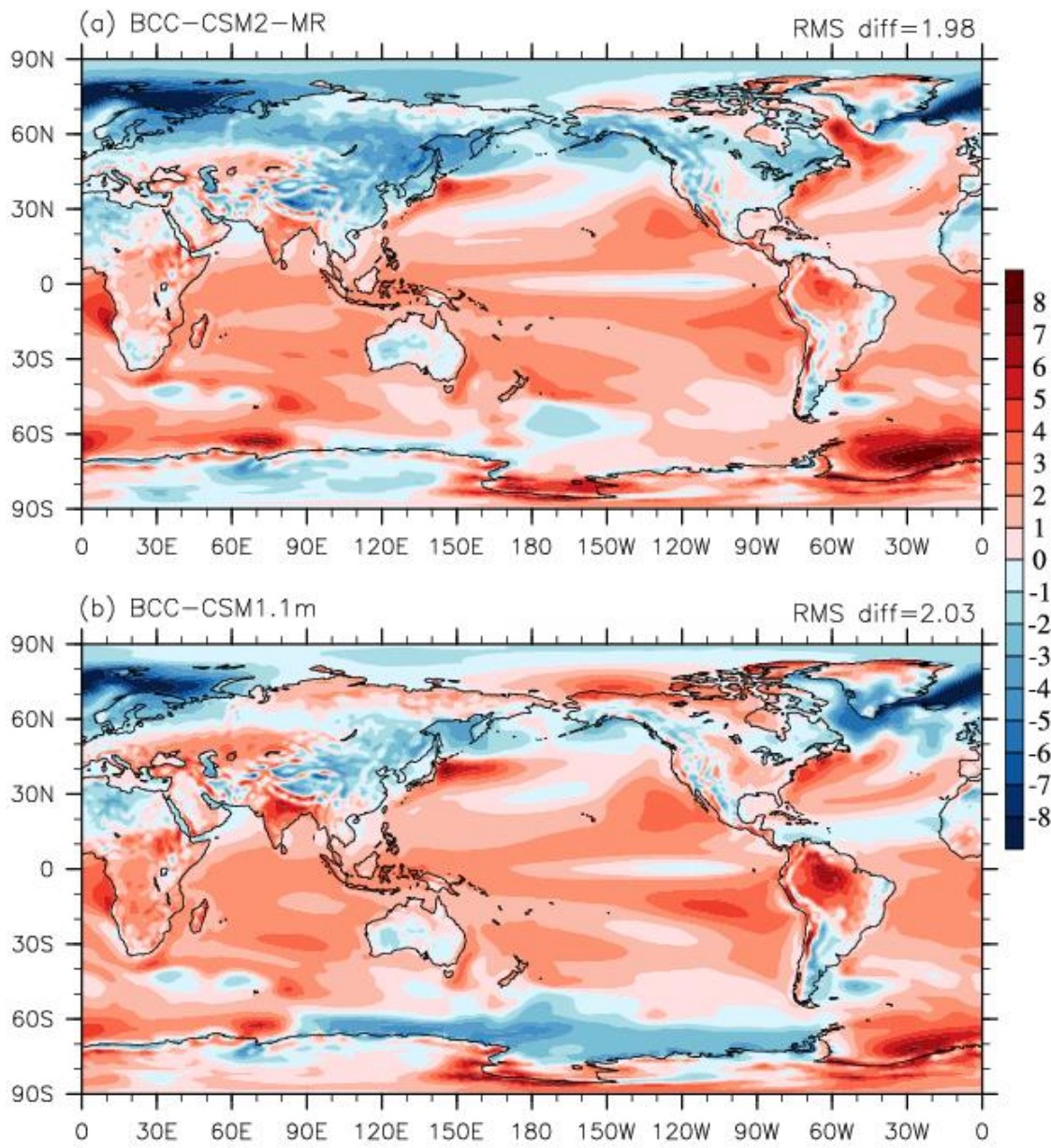

Figure 5. Annual-mean surface (2 meter) air temperature biases (°C) of (a) BCC-CSM2-MR and (b) BCC-CSM1.1m simulations with contrast to the reanalysis ERA-Interim for the period of 1986 to 2005.

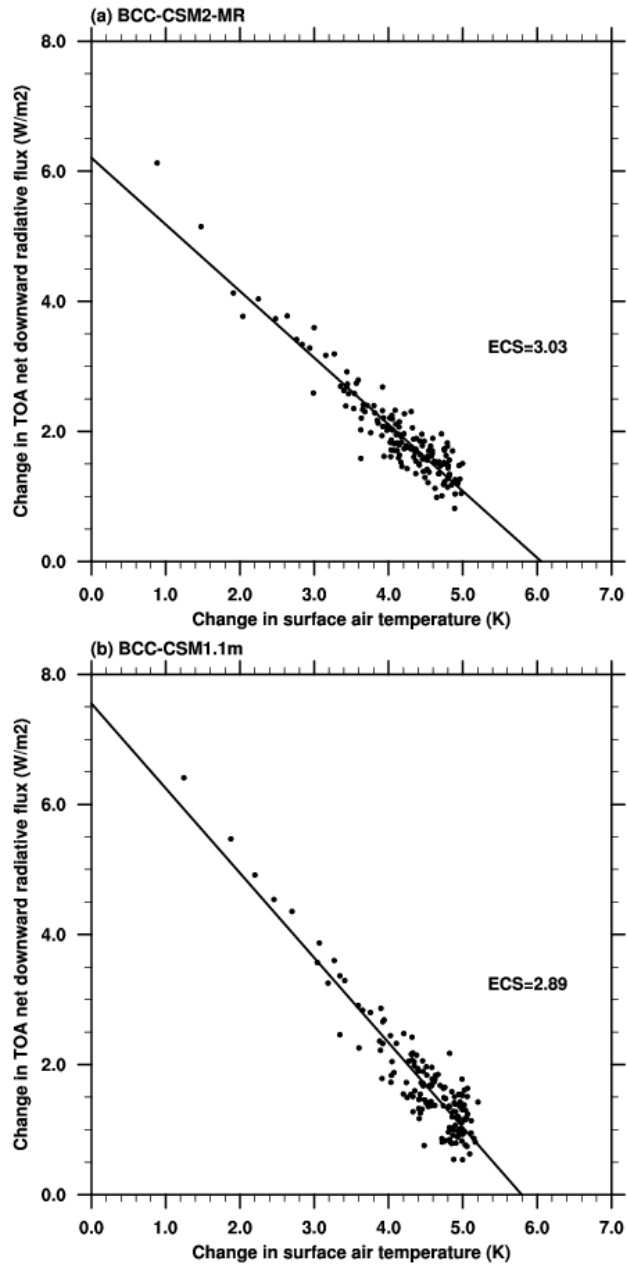

Figure 6. Relationships between the change in net top-of-atmosphere radiative flux and global-mean surface air temperature change simulated with an abrupt 4xCO$_2$ increase relative to the pre-industrial control run.

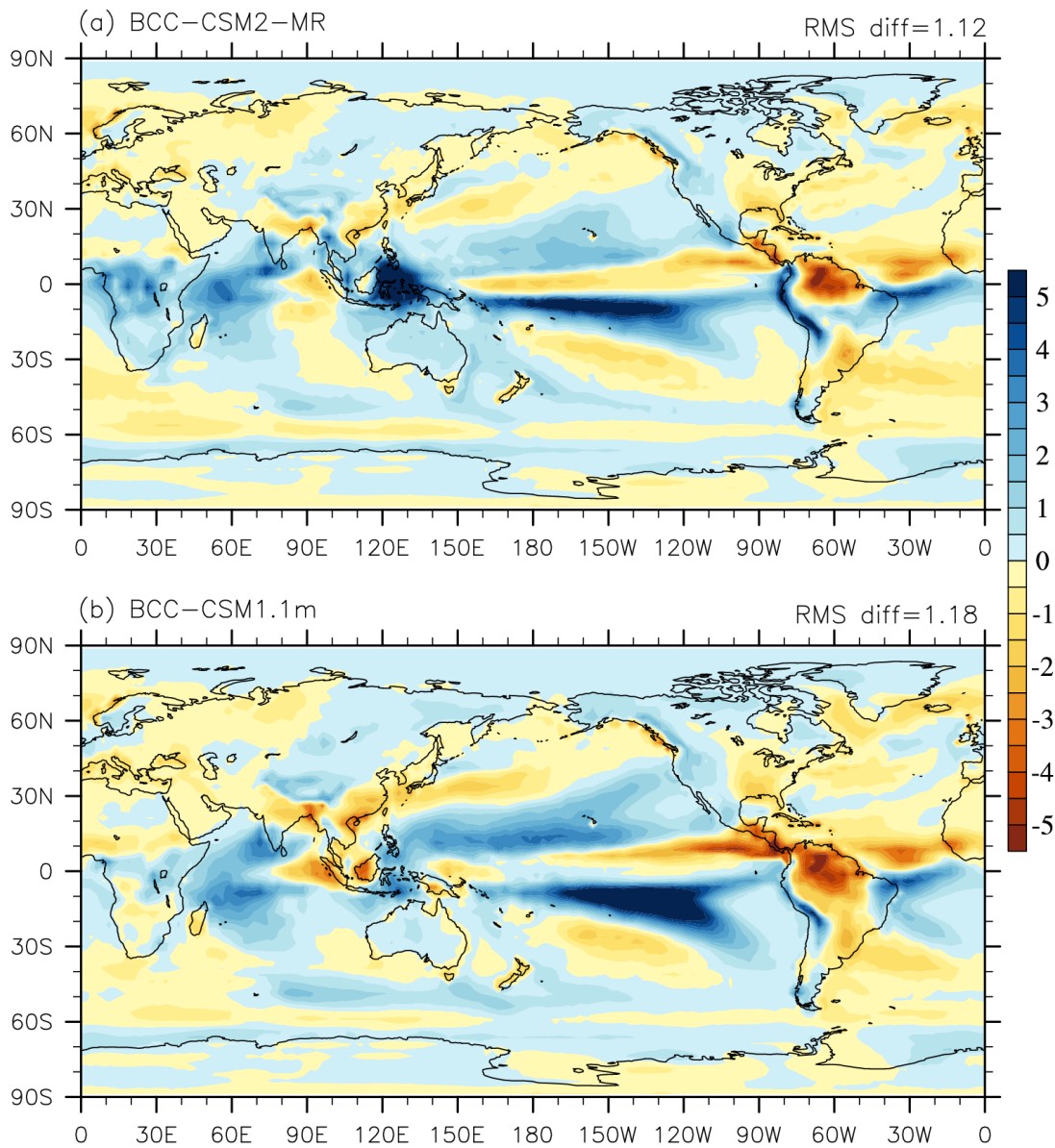

Figure 7. Annual-mean precipitation rate biases (mm day$^{-1}$) of (a) BCC-CSM2-MR and (b) BCC-CSM1.1m simulations with contrast to 1986-2005 precipitation analyses from the Global Precipitation Climatology Project (Adler et al., 2003)

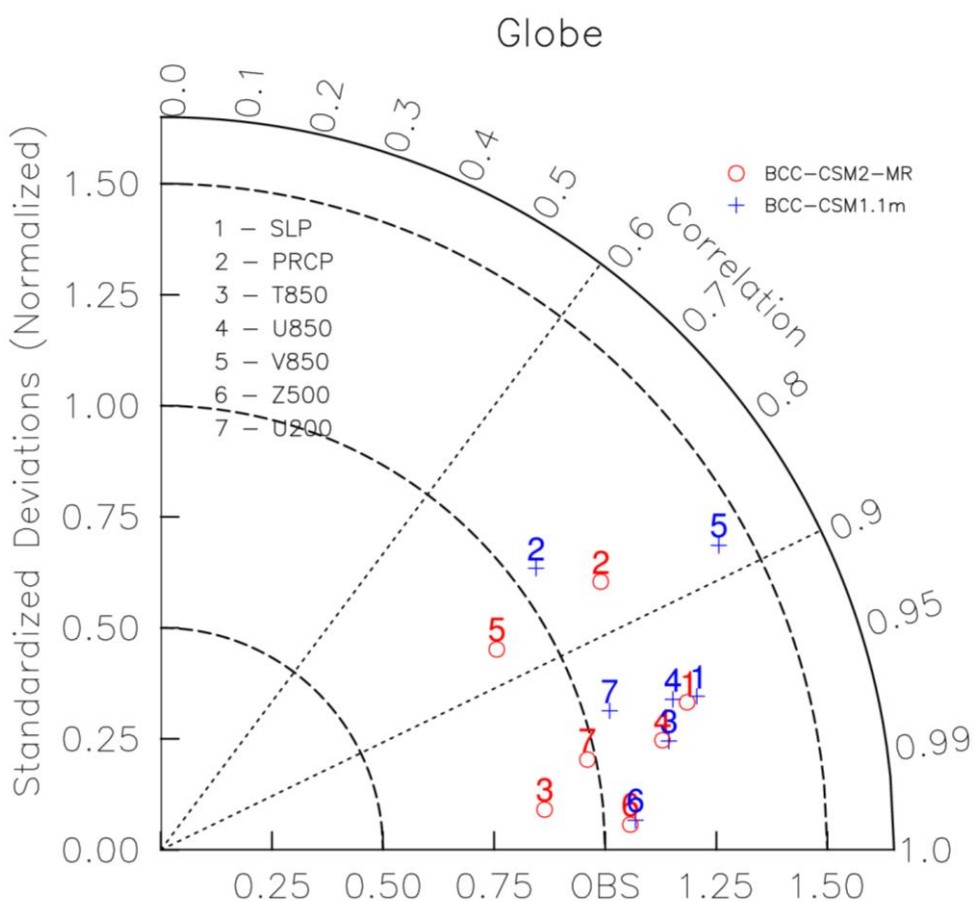

Figure 8. Taylor diagram for the global climatology (1980–2005) of sea level pressure (SLP), precipitation (PRCP), temperature at 850 hPa (T850), zonal wind at 850 hPa (U850), longitudinal wind at 850 hPa (V850), geopotential height at 500 hPa (Z500), and zonal wind at 200 hPa (U200). The radial coordinate shows the standard deviation of the spatial pattern, normalized by the observed standard deviation. The azimuthal variable shows the correlation of the modelled spatial pattern with the observed spatial pattern. Analysis is for the whole globe. The reference dataset is ERA-Interim except the precipitation from Global Precipitation Climatology Project dataset. The model results of BCC-CSM2-MR and BCC-CSM1.1m are the mean for 1980 to 2000. Blue crosses are for BCC-CSM1.1m and circles for BCC-CSM2-MR.

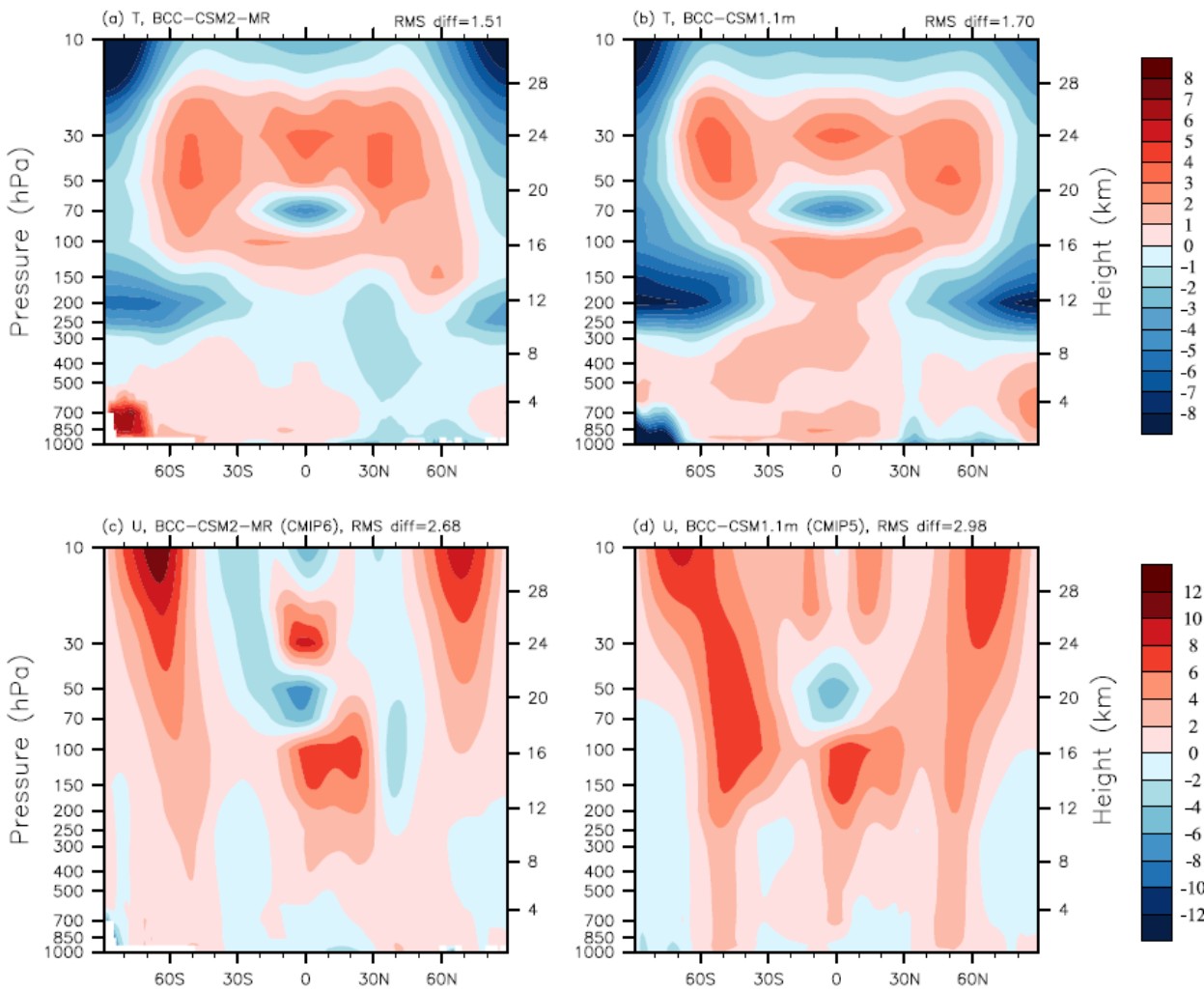

Figure 9. Pressure-latitude sections of annual mean temperature (top panels, K) and zonal wind (bottom, m s$^{-1}$) biases for BCC-CSM2-MR (left) and BCC-CSM1.1m (right), with respect to the reanalysis ERA-Interim for the period of 1986 to 2005.

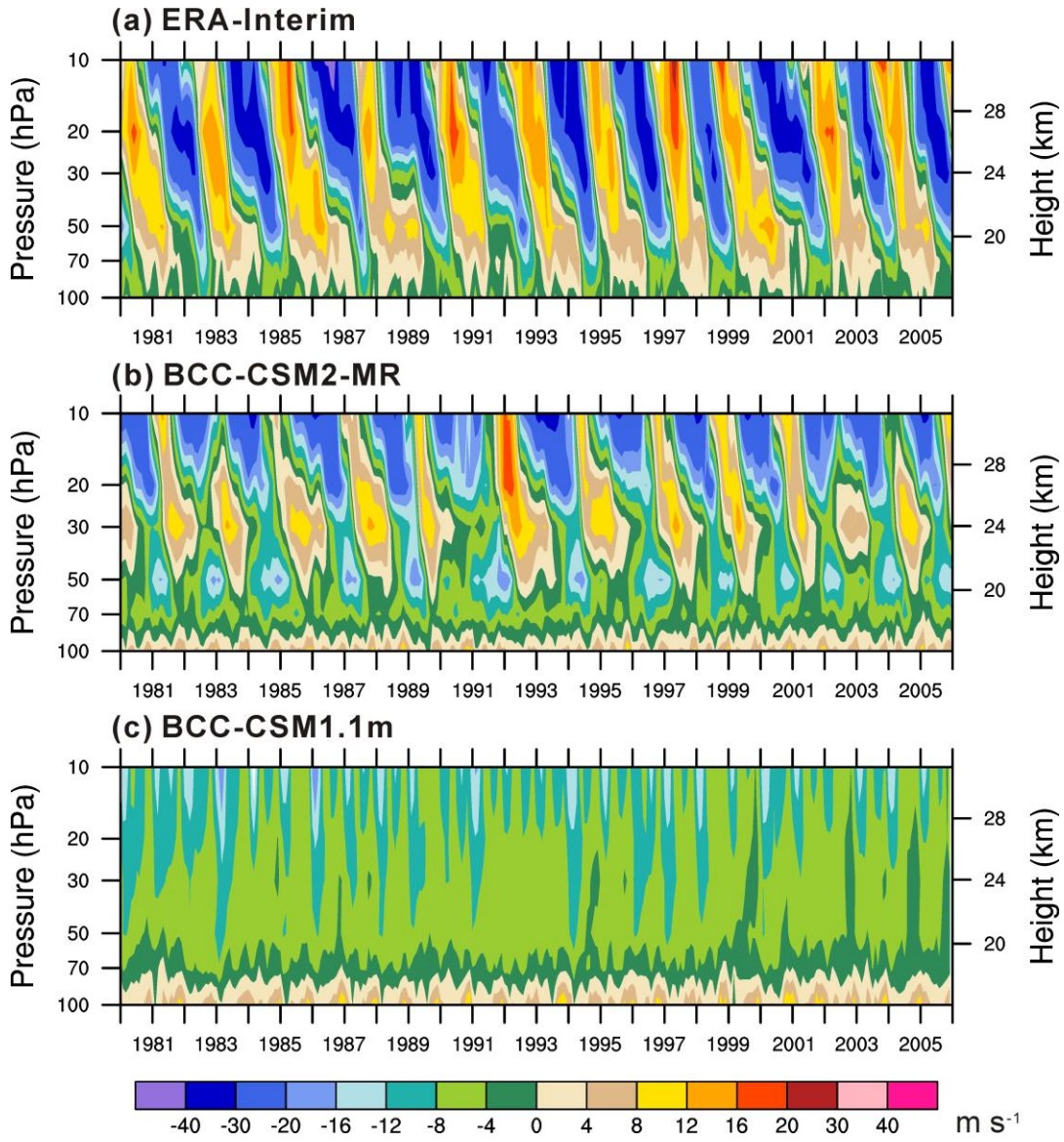

Figure 10. Tropical zonal winds (m s$^{-1}$) between 5°S and 5°N in the lower stratosphere from 1980 to 2005 for (a) ERA-Interim reanalysis, (b) BCC-CSM2-MR, and (c) BCC-CSM1.1m.

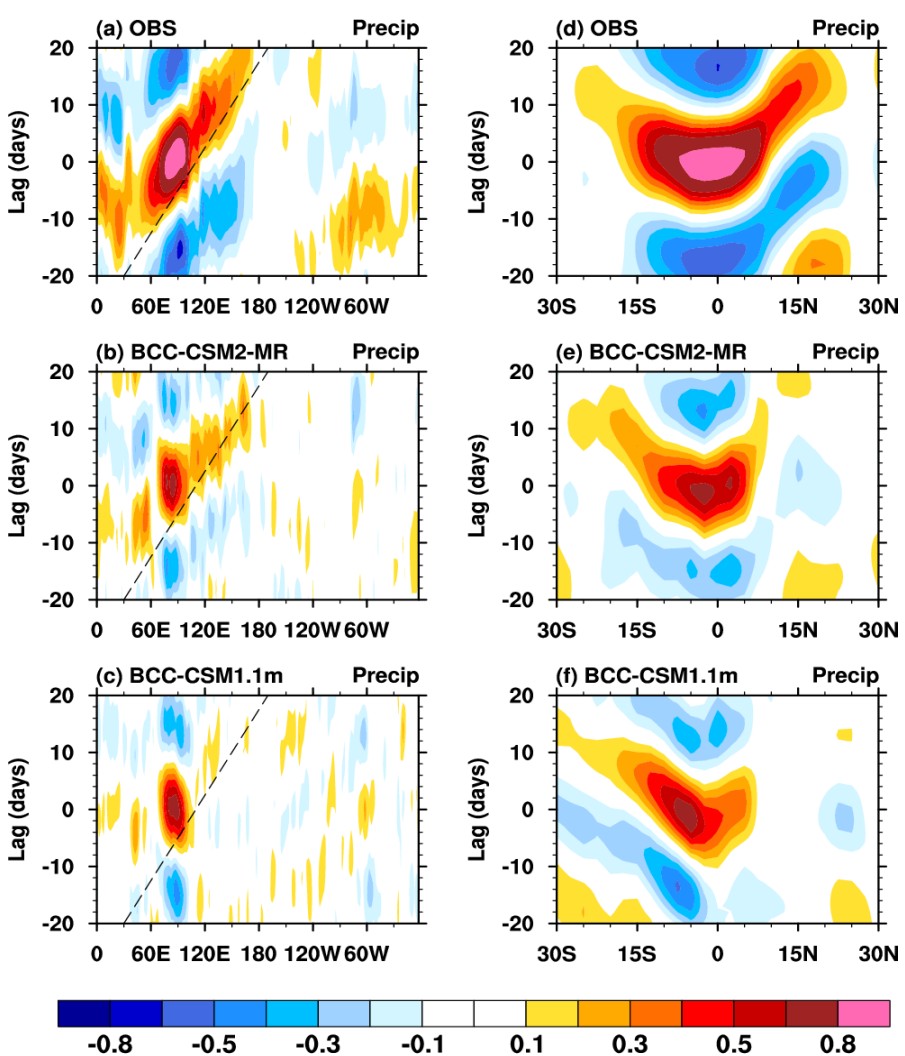

Figure 11. Left panels: longitude-time evolution of lagged correlation coefficient for the 20–100 day band-pass-filtered anomalous rainfall (averaged over 10 °S–10 °N) against itself averaged over the equatorial eastern Indian Ocean (75 °–85 °E; 5 °S–5 °N). Right panels: same as in the left panels but to show meridional propagation of the filtered rainfalls, and lagged correlation coefficient for anomalous rainfall (averaged over 80 °–100 °E) against the rainfall averaged over the same region of equatorial eastern Indian Ocean. Dashed lines in each panel denote the 5 m s$^{-1}$ eastward propagation speed. The reference GPCP observations and historical simulations of models are from the period of 1997-2005.

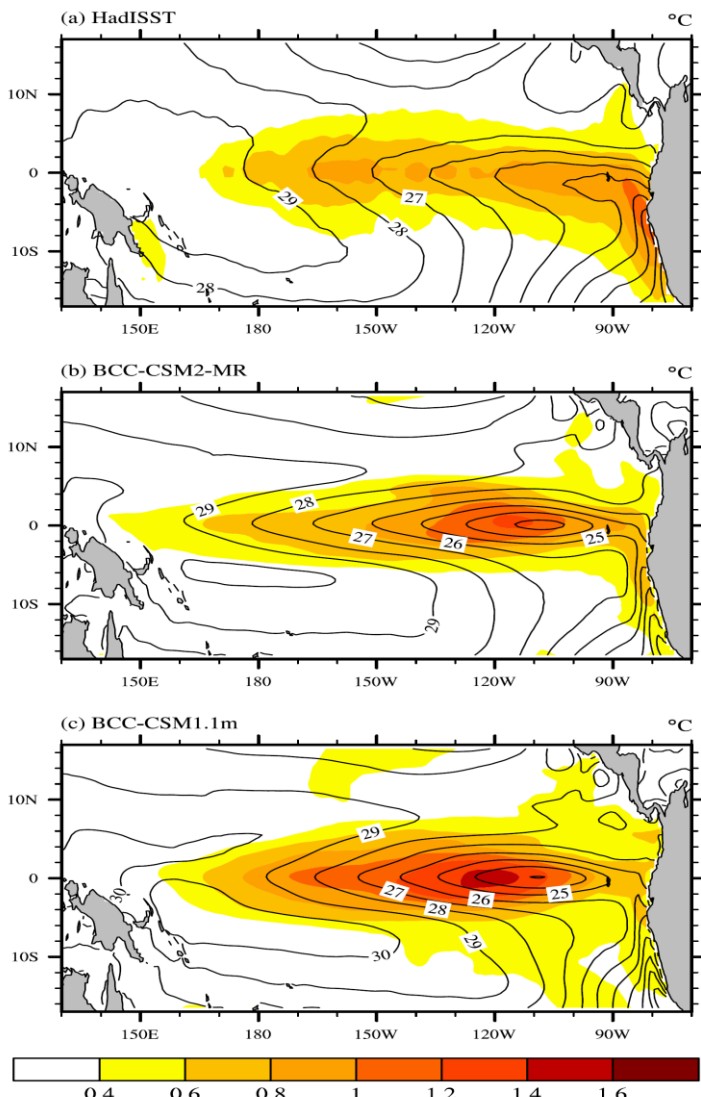

Figure 12. The spatial distributions of 1986-2005 annual mean sea surface temperature (contour lines, ℃) and its standard deviation of interannual anomalies (shaded area, ℃) in the tropical Pacific for (a) HadISST observations (Rayner et al., 2003), (b) BCC-CSM2-MR, (c) BCC-CSM1.1m.

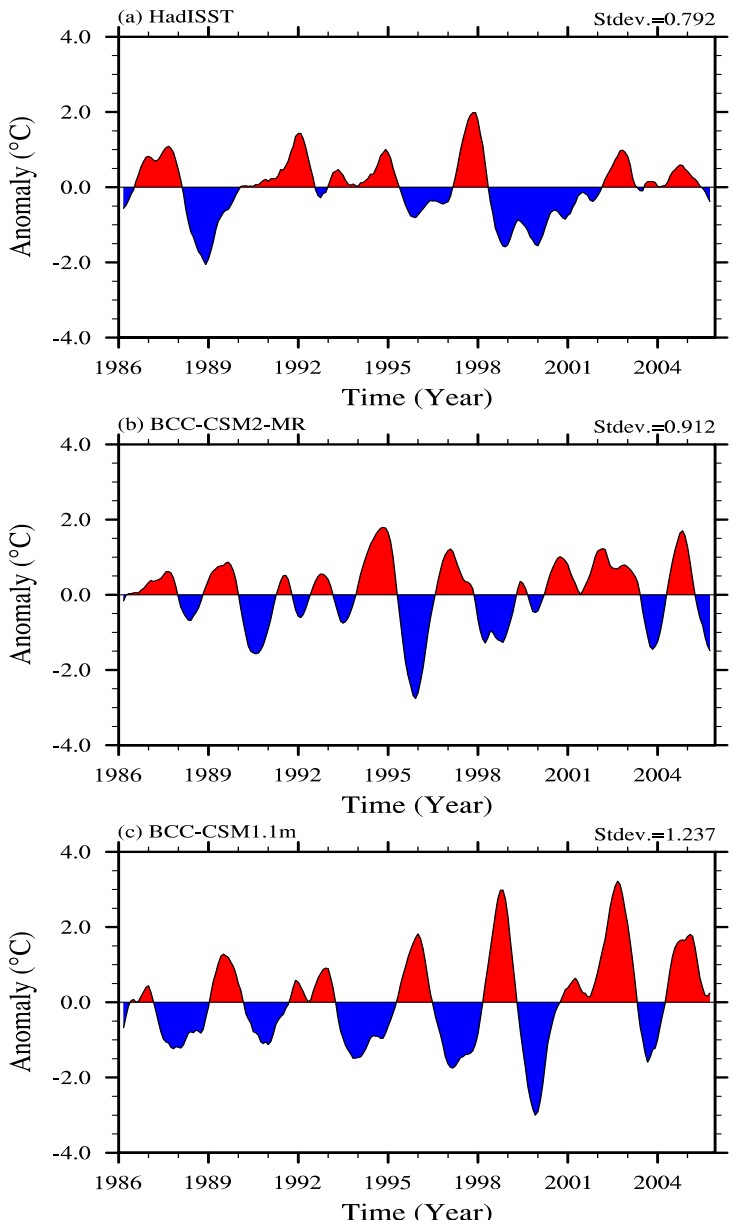

Figure 13. The time series of Nino3.4 SST Index from 1986 to 2005 for (a) HadISST data, (b) BCC-CSM2-MR, (c) BCC-CSM1.1m.

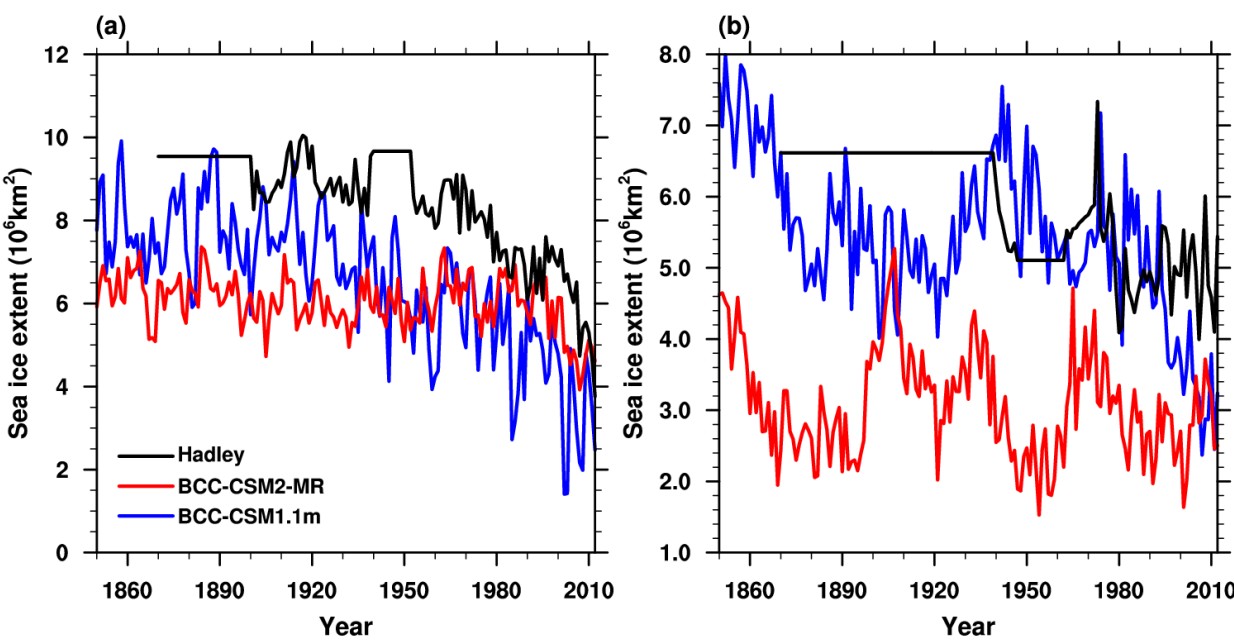

Figure 14. Time-series of sea-ice extent from 1851 to 2012 for (a) the Arctic in September and (b) the Antarctic in March as simulated in BCC-CSM2-MR and BCC-CSM1.1m and observations that are derived from Hadley Centre Sea Ice and Sea Surface Temperature data set (Rayner et al., 2003).

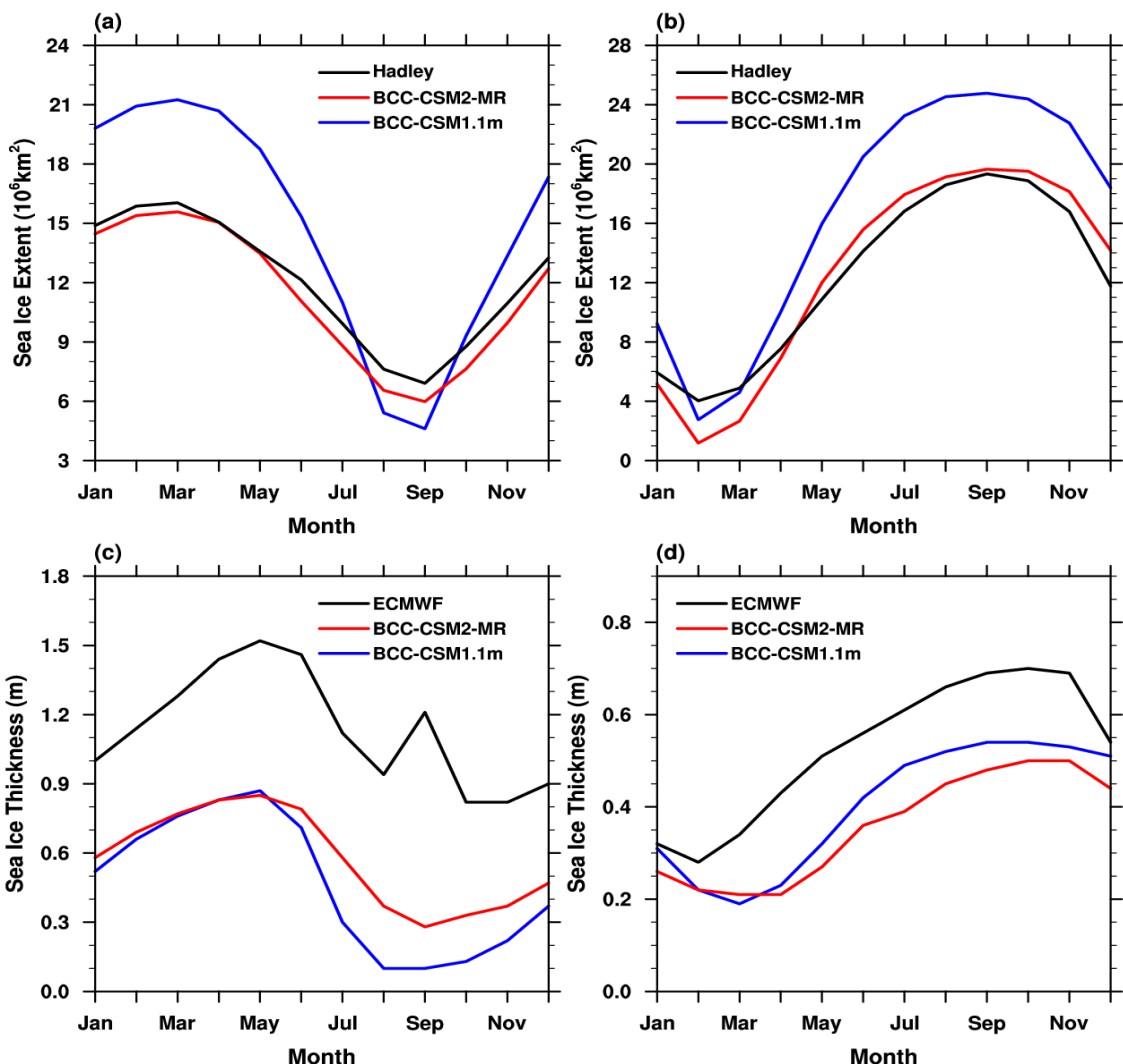

Figure 15. Mean (1980–2005) seasonal cycle of sea-ice extent (upper panel, the ocean area with a sea-ice concentration of at least 15%) and mean thickness (lower panel) in the Northern Hemisphere (left) and the Southern Hemisphere (right). The observed seasonal cycles of sea-ice extent in (a) and (b) are derived from 1980-2005 Hadley Centre Sea Ice and Sea Surface Temperature data set (Rayner et al., 2003), and the ice thickness in (c) and (d) are derived from 1980-2005 global gridded data set based on European Center for Medium-Range Weather Forecast (Tietsche, et al., 2014).

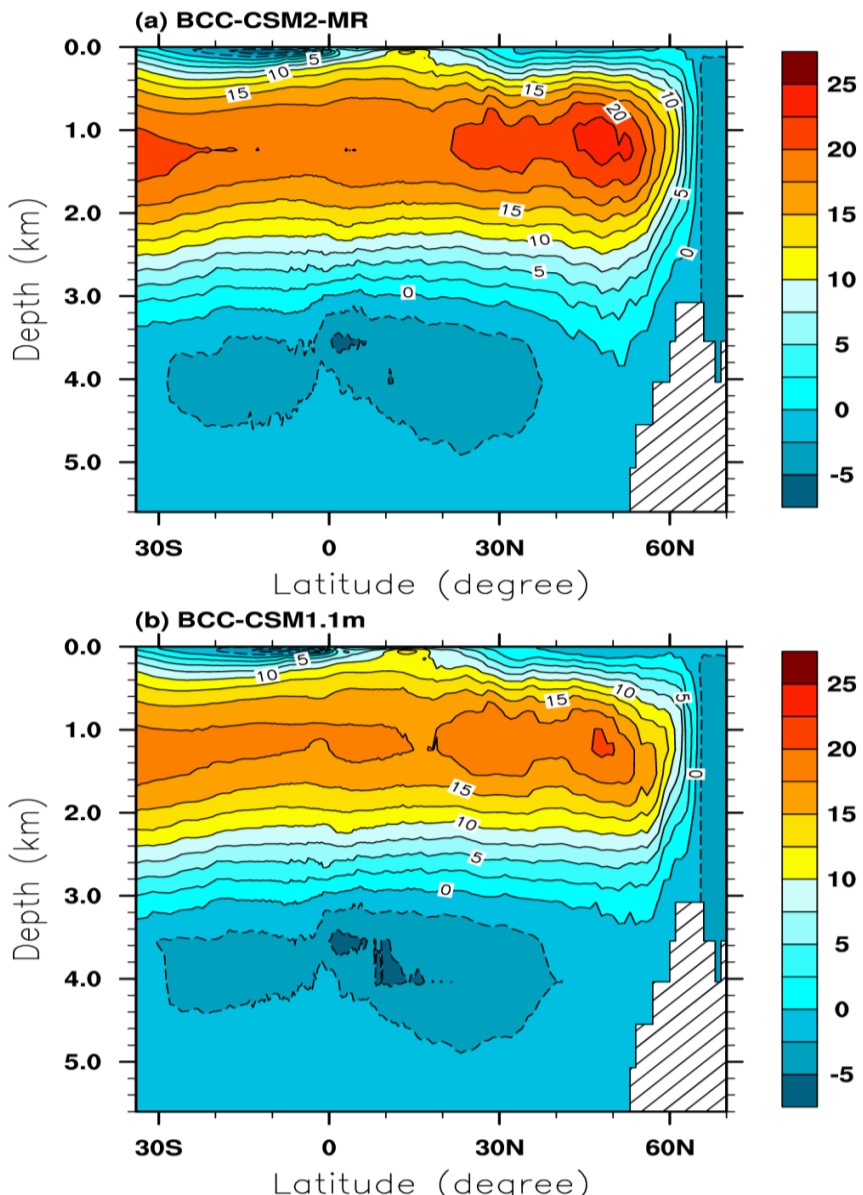

Figure 16. Zonally-averaged streamfunction of the Atlantic Meridional Overturning Circulation (AMOC) for the period of 1980 to 2005 in BCC-CSM2-MR (top) and BCC-CSM1.1m (bottom). Units: Sv

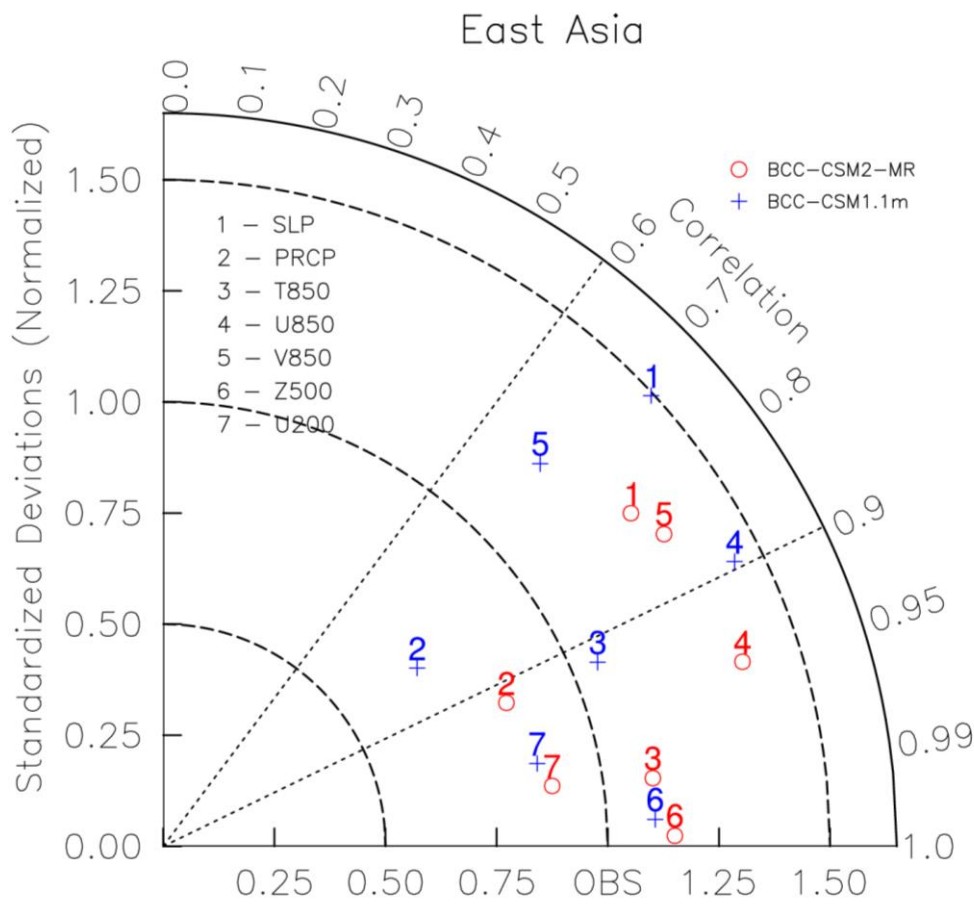

Figure 17. Same as in Figure 8, but for the domain covering East Asia (20°-50°N, 100°-140°E).

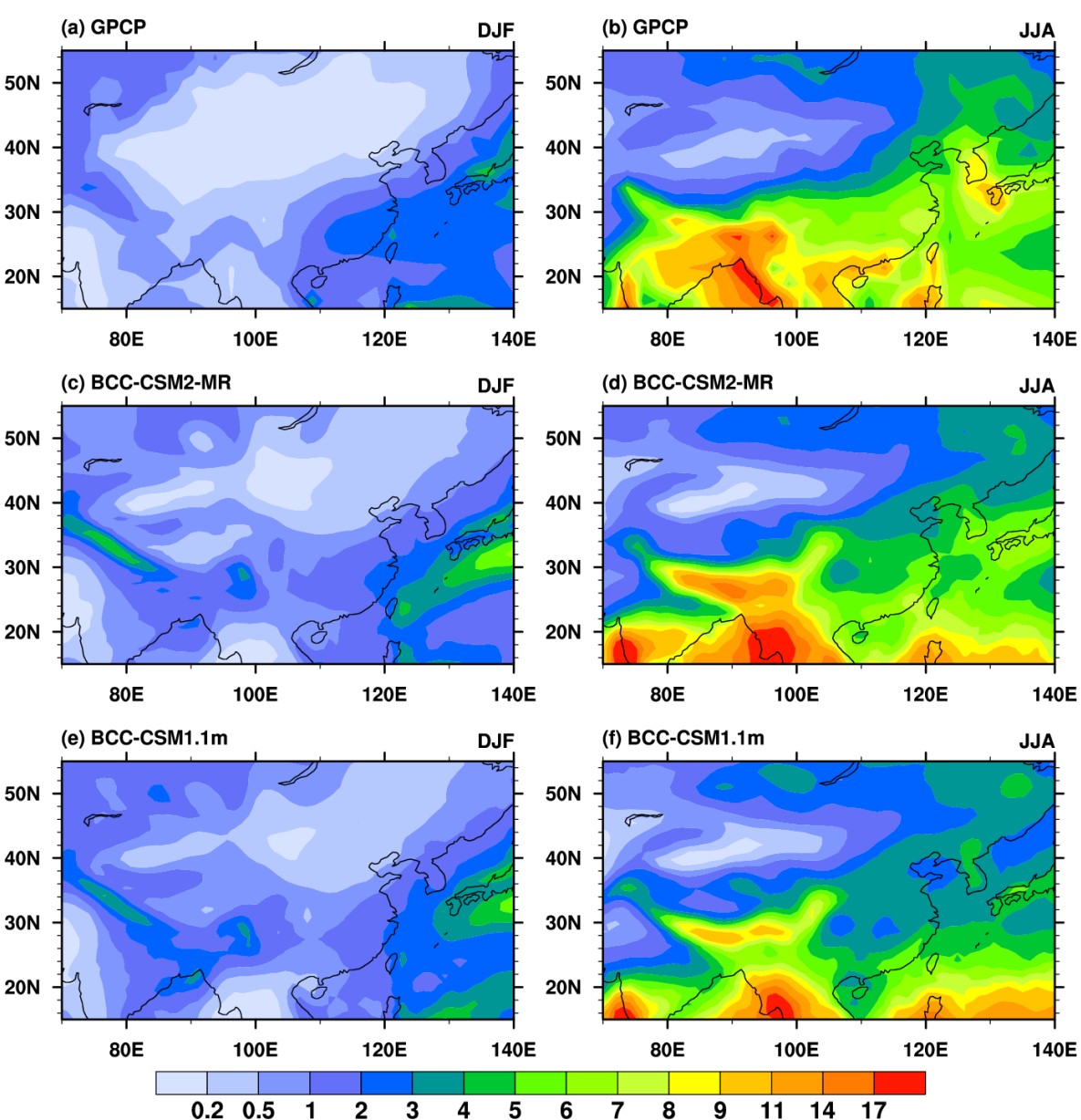

Figure 18. Regional distribution maps of precipitation climatology (averaged from 1980 to 2005) for December-January-February (left panels) and June-July-August (right panels) from (a) GPCP, (b) BCC-CSM2-MR, (c) BCC-CSM1.1m. Units: mm day$^{-1}$.

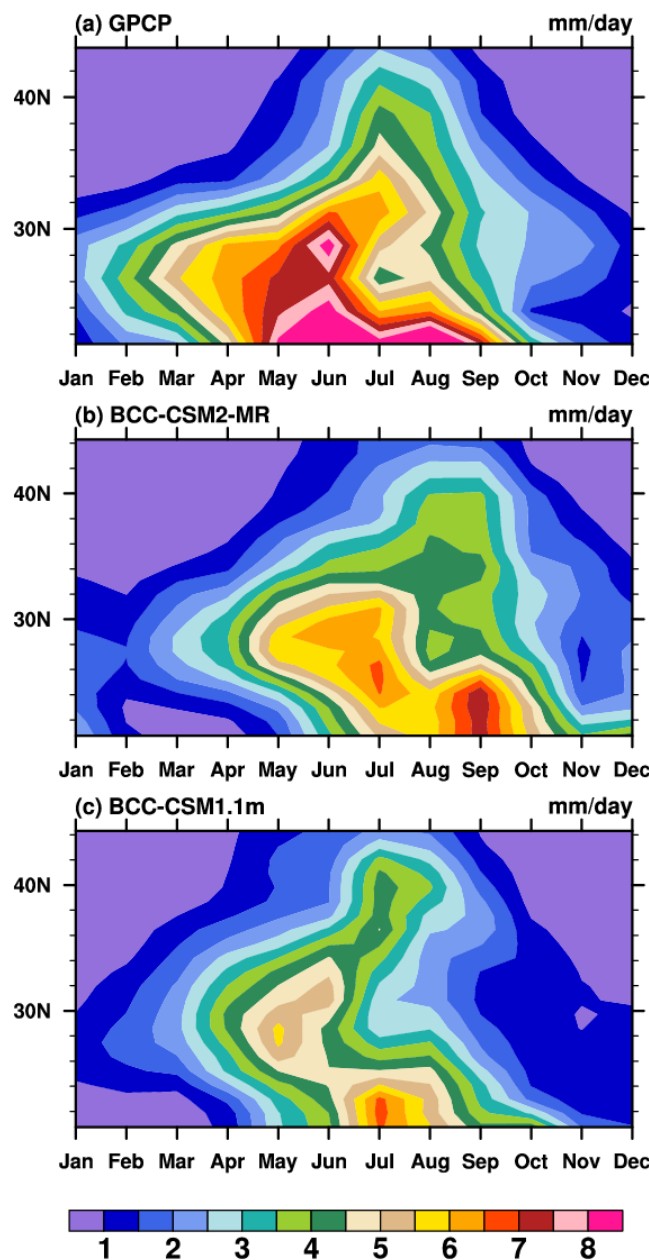

Figure 19. Latitude (from 20°N to 25°N) – month (Jan. to Dec.) diagrams showing variations of monthly precipitation averaged over 100°-120°E and for the period of 1980-2005. (a) GPCP, (b) BCC-CSM2-MR, (c) BCC-CSM1.1m. Units: mm day$^{-1}$.

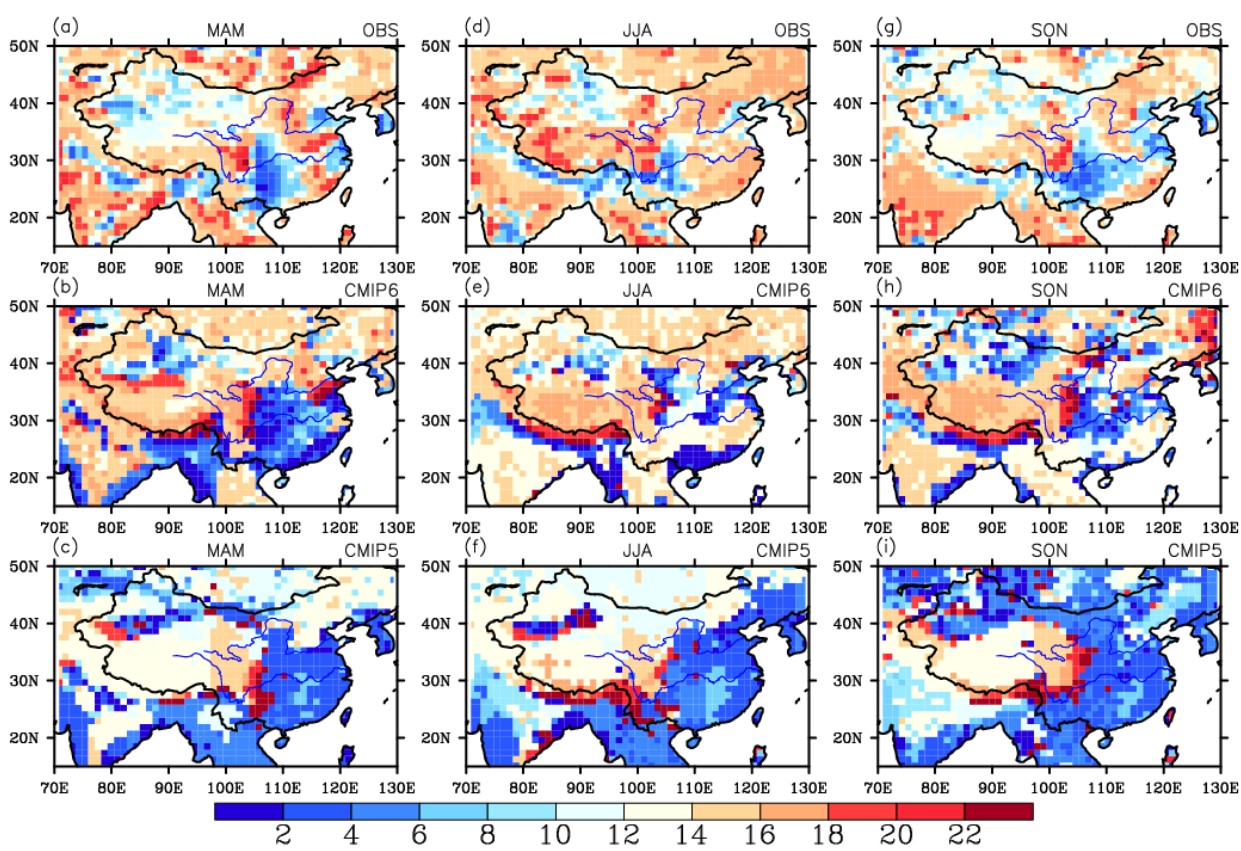

Figure 20. Local times of maximum frequency of rainfall occurrence in March-April-May (left column), June-July-August (middle column), and September-October-November (right column) over China and its surrounding areas for BCC-CSM2-MR (middle panel), BCC-CSM1.1m (bottom panel), and TRMM data (top panel, Huffman et al., 2014). The rainfall occurrence is defined as the hourly precipitation larger than 1 mm.