# Peer review of "The Beijing Climate Center Climate System Model (BCC-CSM): Main"

_Geoscientific Model Development, 2018_

## Referee Comment (RC1) · Anonymous Referee #1 · 13 Jan 2019

General comments:

In this article the authors describe the changes from the BCC-CSM1.1 model that was used in CMIP5 to the new BCC-CSM2 model that is now employed in CMIP6. They compare the "historical" simulations as specified by CMIP5 and CMIP6, respectively, of the old and new model. They show that the new model can simulate the mean climate as well as some modes of variability with some skill, and they point out a number of improvements in the new simulation compared to the old one. Obviously this article is meant to be a basic, citable documentation for the new CMIP6 simulations of the BCC-CSM2 model.

[Figure]

The documentation and analysis of the simulations is superficial. No attempt is taken to describe or analyze the effects of the single model improvements, or to attribute the identified differences between the two simulations to the introduced model changes. Therefore there is not much that the reader can learn from this article, except that the coupled model can reproduce the transient climate of the CMIP historical experiment with some skill. The authors point at future publications ("to be submitted") for other experiments, model resolutions or the QBO in the historical simulations.

Detailed comments:

L40: ... Many climate models in the world have been developed since the IPCC-AR4, ... The IPCC reports certainly motivated many groups to contribute climate projections for the assessment of future climates, for which these groups developed suitable models. But I do not see why IPCC-AR4 is pointed out as a special landmark along the path of the development of global climate models or Earth system models. Rather the initial coupled model inter-comparison project (CMIP) would deserve to be mentioned, and the way the CMIPs were developed by the community and the working group on coupled modeling (WGCM).

L71: ... Its performance is presented in a separated paper (Wu et al., to be submitted). ... Please check if references to "to be submitted" articles are allowed in GMDD. Better write "... Its performance will be presented in a separate paper. ...".

L78: Please include a figure that compares the profiles of layer thickness against the height of the layer for the L26 and L46 grids, using for example a simple log-p height definition. This would clarify how the vertical resolution has been improved.

L81: 2.1 Atmospheric component BCC-AGCM This is the main documentation for the new version of the atmospheric model. Therefore I would expect to find here basic information for all processes. This should include the numerical techniques used in the dynamics, the transport scheme and the physics. This basic information should be kept concise, so that most of the room can still be devoted to the subsections (a) to (d)

for the changes compared to the preceding documented version.

L96: . . . considered . . . → . . . considers . . .

L100: . . . environment, The mass . . . → . . . environment, the mass . . . L128: . . . at each model grid . . . → . . . in each model grid cell . . . ? L138: . . . T_con → T_conv . . .

L144: ". . . Following the method above, the cloud fraction (C_deep and C_shallow), temperature (T_deep and T_shallow), specific humidity (q_deep and q_shallow) for the deep convective, shallow convective clouds can be then deduced sequentially. . . . " Does the scheme allow the concurrent occurrence of shallow and deep convection in the same atmospheric column, or is only one type of convection allowed at any one time in a single column? If concurrent occurrence is allowed, how is the parametrized computation of deep and shallow convection split, and how is the necessarily sequential diagnostics of C, T and q in the shallow and deep updrafts organized?

L151 and L153: These equations can be numerically unstable in the limit of C_deep+C_shallow→1, because of the division by (1-Cdeep-Cshallow) that is needed to obtain the unknowns T_ambient and q_ambient. Is this a practical problem, or rendered irrelevant by the multiplication factor (1-C_conv) in Eq. 2?

L157: . . . RH_abmient . . . → . . . RH_ambient . . .

L164: I cannot find the publication by 'Kristj ÌĄansson and Kristiansen [2000]'. Do you mean 'Kristjánsson et al., 2000'? Kristjánsson, J. E., J. M. Edwards, and D. L. Mitchell (2000), Impact of a new scheme for optical properties of ice crystals on climates of two GCMs, J. Geophys. Res., 105(D8), 10063–10079, doi:10.1029/2000JD900015.

L202: . . . k = 1.18 × 106 cmˆ−1 secˆ−1 . . . → . . . k = 1.18 × 10ˆ6 cmˆ−1 secˆ−1 . . .

L206: Section 'd. Parameterization of gravity wave drag' This paragraph discusses drag by dissipating gravity waves originating from flow over orography or atmospheric sources. What about drag by blocking effects from unresolved orography? Are such effects, which sometimes are included in gravity wave parameterizations, considered

in BCC-AGCM3-MR?

L221: …. This parameterization scheme of convective gravity waves can improve the model's ability to simulate the stratospheric quasi-biennial oscillation in BCC-AGCM3-MR. … This is a rather general statement. It is clear that non-orographic gravity waves make a significant contribution to the forcing of the QBO, and if the gravity waves are not resolved, then their effect needs to be parameterized for the simulation of the QBO, and tunable parameters can be used to improve the structure of the QBO. Is the CF parameter, which you tune, valid for all latitudes or only for equatorial latitudes, where the QBO exists?

L277: Is the "simple scheme about the surface albedo, roughness length, turbulent sensible and latent heat fluxes over rice paddies" documented, or is there a manuscript in preparation? If not, and if the scheme is indeed simple, you should include the documentation here.

L348: … The preindustrial climate state of BCC-CSM2-MR is preceded by a more than 500 years piControl simulation following the requirement of CMIP6. … Which were the goals of the spin-up simulation for the piControl experiment? Which were the criterions for declaring the spin-up phase completed? It would be interesting to learn about these criterions.

L.353: … the up-limit of the atmosphere … → … the top of the model atmosphere …

L.362: … It means that the whole earth system in our models is very close to energy equilibrium. … For a transient period it is a bit difficult to judge from the similarity between TOA radiation fluxes of the model and the observations whether the model is generally in a "good" equilibrium. How is the energy balance for the preindustrial control experiment? Here we know that the net energy flux at TOA should be zero except for fluctuations related to the internal variability of the coupled system. If the model system has energy leaks, as many climate models have, we should see this clearly in the stabilized piControl simulation. Such a leakage would have to be considered in the

comparison of the transient TOA radiation fluxes of the model and the observations. My suggestion is that you discuss the net energy flux at TOA and the surface and maybe other quantities of interest of the piControl simulation before starting the discussion of the historical simulation. This could for instance be embedded in a new section that explains the tuning goals of the piControl experiment.

L372: ... These biases are reduced in BCC-CSM2-MR. ... Which of the model changes discussed earlier cause the strong (and useful) changes in the tropical SW and LW cloud radiative forcing between the old and new model?

L379: ... Biases of annual mean surface air temperature (at 2 meters) ... Figure 2 shows the spatial patterns of the T2m bias. But first of all I am wondering how the transient global mean temperature is evolving from the stabilized pre-industrial mean temperature representative for 1850 to the present day. Please discuss first the global mean evolution before describing the pattern of the T2m bias near the end of the historical experiment. Further, it would be interesting to read your opinion on the contribution of the model changes to the observed differences between the models. Can you attribute the disappearance of the cold bias in the southern oceans in the new model compared to the old model to the changes in the model for turbulent fluxes over sea ice? Are specific changes of the land model important for the increased cold bias in east Asia and Siberia in the new model compared to the old model? For the land surface biases it would be valuable to know if these patterns are already present in AMIP simulations, where effects of oceanic biases are excluded.

L386: annual mean precipitation What is the global mean precipitation in both models? Do you have any thoughts about the contribution of the changes in the deep convection scheme to the strong wet bias in the Maritime Continent?

L. 393: ... The evaluation is done against climatology of ERA-Interim ... NCEP dataset ... From the text and the figure caption it seems rather that only NCEP is used and not ERA-Interim. Can you please clarify this?

[Figure]

L421: . . . Given a much higher vertical resolution and an advanced parameterization of the gravity wave drag . . . Despite the more complete parameterization of gravity wave drag – now including gravity waves from atmospheric sources – the zonal mean zonal wind biases in the high latitudes of the stratosphere have increased near 10 hPa, where one would expect the main benefit from gravity wave parameterizations. Can you explain why there is no benefit from the improved GWD parameterization on the structure of the polar night jets? Did you attempt to tune the gravity wave drag to reduce errors in the stratospheric extratropical zonal circulation?

L427: . . . In Figure 6(b), the BCC-CSM2-MR simulations present a clear quasi-biennial 428 oscillation of the zonal winds as observed. . . . The downward propagation . . . does not penetrate to sufficiently low altitudes. . . . Though the vertical resolution is increased, it is still too low to expect a QBO simulation down to at least 70 hPa, because the forcing from resolved waves cannot be adequately resolved. Therefore it seems like the QBO occurring in the new model must be dependent entirely or nearly entirely on the Beres parameterization. Has this scheme been tuned to obtain the QBO in the new model?

L470: . . . The most remarkable improvements of BCC-CSM2-MR appear in the boreal warm seasons, . . . To which model improvement do you attribute the strong improvement of SIE or SIC?

L495: . . . Our CMIP6 model can capture this warming hiatus. . . . The word "capture" suggests that the hiatus is a predictable climate feature that a coupled climate simulation can be expected to reproduce if the forcing is realistic and the model is "correct". Is this what you want to express? Maybe it is better to write for instance: "The historical simulation of the CMIP6 model shows a hiatus towards the end of the simulation that resembles the observed one." Do you have other ensemble members for the historical CMIP6 simulation, and if so do all members reproduce the hiatus of 1998-2013? The figure shows also that the CMIP5 simulation is significantly colder in the early decades than observations or the CMIP6 simulation. Later on, however, both simulations evolve

by and large in a similar way. Can this be explained by the external forcing (volcanic aerosols?)? Do you have any insight that you can share?

L509: . . . Observation-based NSIDC data are also plotted when available. . . . The caption for Fig. 11 reads: ". . . observations-based Hadley Centre Sea Ice and Sea Surface Temperature data set (Rayner et al., 2003)." Please clarify.

L516: 4.5 Climate sensitivity to CO2 increasing Figure 12 following Gregory (2012) not only provides estimates for the ECS, but primarily provides information on the climate feedback. Comparing both models, the ECS is similar, but the feedback parameter is substantially different: BCC-CSM3-MR: ca. -1 W/m2/K; BCC-CSM1.1m: ca. -1.3 W/m2/K Thus the result that both ECS values are very similar results only because the initial 4xCO2 forcing is quite different: BCC-CSM3-MR: ca. 6 W/m2; BCC-CSM1.1m: ca. 7.5 W/m2/K Can you please comment on the origin of the large difference in the initial forcing?

L529: . . . abruptCO2 . . . → . . . abrupt4xCO2 . . . ?

---

## Referee Comment (RC2) · Anonymous Referee #2 · 17 Jan 2019

This is a description paper of BCC-CSM2-MR. The upgraded schemes are well described, while evaluations for internal variability and long-term trends are insufficient. Authors emphasized improvements in several aspects. Some of them (e.g., representation of QBO) are remarkable improvements but some others are not very convincing. Major revisions and some additional analyses are required as follows:

Major comments:

The paper lacks the information of ensemble size of each historical runs. The following two comments are related to this issue:

1.1 Is the global warming trend of CMIP6 model shown in Fig. 10 the ensemble-mean value? What is the gray shading? Maximums/minimums of CMIP5? If the ensemble size of CMIP6 is only one member, you cannot deny the possibility that the simulated slowdown after 1998 happened by chance due to natural variability. On the other hand, if it is ensemble-mean value, then the slowdown after 1998 implies it is the result of external forcing. In the latter case, you need to discuss whether the difference in the trends between CMIP5 and CMIP6 is due to the difference in external forcing or the model update.

1.2 Is the improvement of MJO shown in Fig. 7 robust? A correlation coefficient depends on a sample size. Are the sample sizes (the total numbers of years) you used for the analyses of the observation, BCC-CLM1.1m, and BCC-CLM2-MR comparable? If the improvement is true, what do you think is a factor improving the representation of MJO? New convection scheme?

It seems that the results in Fig. 8 and Fig. 11 are inconsistent. Is it due to the difference between the reference datasets (NSIDC, ECMWF forecasts, and Hadley Centre Sea Ice)? Please add some explanation to figure out it and the reason why you used those datasets.

Section 4.2: Please discuss the SST cold bias in the equatorial Pacific and the double ITCZ problem. Evaluations for the subsurface ocean (temperature, zonal currents) are also required.

Section 4.6: There is a large seasonal cycle in the East Asian climate. Analyses should be done for each season, especially for JJA and DJF.

Comparison of ENSO representation (NINO3.4 time series, amplitude, spatial pattern etc.) is necessary to evaluate the model performance.

Minor issues:

L19: models -> model's

L27-28 "Compared to BCC CMIP5 models, BCC CMIP6 models show. . .": The expression "models" is inappropriate. This paper compared only one model for each (BCC-CSM1.1m for CMIP5 and BCC-CSM2-MR for CMIP6).

L41: More -> more

L53: The full name of CMIP5 and Taylor et al. (2012) should appear at L42.

L59: Please comment on Section 5 and 6.

L64: Please add the full name for NCAR.

L64: Coupler -> coupler

L65-66: Are tuning parameters also the exactly same between the two models?

L79: Please add the level of the top layer.

L240: What is the CEVSA model?

L243: Table 2 -> Table 3

L362: 2014 -> 2005? The end of the CMIP5 historical run is 2005.

L371 "1986-": "1985-" in the caption of Fig. 1

L374-376: In the mid latitude, discrepancies in the two observations are also large. You cannot discuss the difference between the two model.

L396: Precipitation data diagnosed in reanalysis is not necessarily correct. You had better use the observation-based dataset.

L406: Figure 4s -> Figure 4

L409: Were there any reason you used the two different reanalysis datasets (NCEP and ERA-interim) for each analysis?

L437 "forcing is less adequate": What forcing?

L452: improvementcompared -> improvement compared

L478-481: Is the lower and deeper NADW better? Please show the observation based values.

L486 "from 60S to 60N": Why did not you use the average from 90S to 90N?

L491, L500: HadCRU -> HadCRUT4

L503: Figure 2c -> Figure 10?

L529: abruptCO2 -> abrupt CO2

L535 "the TCR of the new version model BCC-CSM2-MR is lower than BCC-CSM1.1m,": lower -> higher? It is inconsistent with L527-528.

L566-567: Are there any impact of the upgraded land surface scheme on the improvement of the rainfall diurnal cycle?

L616-620: Please see 31.

Table 3 "(Guenther et al., 2012)": Please check the font.

Table 4: What is the value of the net energy at TOA for OBS? 0.81?

Table 4, Notes, "1981-2014": 1981-2005 for BCC-CSM1.1m? Why are the periods slightly different among the figures or tables (1985-2005 for Fig. 1, 1986-2005 for Fig. 2-3 and 5, 1980-2005 for Fig. 4, 8-9, 13)?

Figure 7 d-f: Please show the range of longitude used for the average to obtain the latitude-time section.

Figure 14: The result of BCC-CSM1.1m (BCC-CSM2-MR) should be in the bottom (middle) as same as in the other figures.
* * *

---

## Author Comment (AC1) · 10 Feb 2019

Dear Referee #1,

We would like to thank you for your comments and suggestions to improve the quality of our manuscript "The Beijing Climate Center Climate System Model (BCC-CSM): Main Progress from CMIP5 to CMIP6" by Tongwen Wu et al. Following your suggestions, we added 5 further figures in the revised manuscript. So, all figures in the first manuscript are renumbered. We have also changed the order of presentation for some subsections in Section 4 (Results). The point-to-point responses to your comments are enclosed in the following.

Best regards,

Tongwen Wu and all co-authors

========================================
**Response to Anonymous Referee #1**

**General comments:**
**In this article the authors describe the changes from the BCC-CSM1.1 model that was used in CMIP5 to the new BCC-CSM2 model that is now employed in CMIP6. They compare the "historical" simulations as specified by CMIP5 and CMIP6, respectively, of the old and new model. They show that the new model can simulate the mean climate as well as some modes of variability with some skill, and they point out a number of improvements in the new simulation compared to the old one. Obviously this article is meant to be a basic, citable documentation for the new CMIP6 simulations of the BCC-CSM2 model.**
**The documentation and analysis of the simulations is superficial. No attempt is taken to describe or analyze the effects of the single model improvements, or to attribute the identified differences between the two simulations to the introduced model changes. Therefore there is not much that the reader can learn from this article, except that the coupled model can reproduce the transient climate of the CMIP historical experiment with some skill. The authors point at future publications ("to be submitted") for other experiments, model resolutions or the QBO in the historical simulations.**

**Response:**
Thanks for the relevant comments. Yes, the purpose of this manuscript is to document the transition of our model BCC-CSM from CMIP5 to CMIP6. We hope that it can be a reference for the different experiments of CMIP6 which are progressively available (or very soon for some specific runs) for the scientific community of CMIP6. So we compare here only the general performance between our old and new models, particular performance being investigated in detail in other specific papers. Nevertheless, we think that it is useful to add some more materials (including 5 new illustrations), which goes to the general sense of both

Reviewers.

**L40: …Many climate models in the world have been developed since the IPCC-AR4 …The IPCC reports certainly motivated many groups to contribute climate projections for the assessment of future climates, for which these groups developed suitable models. But I do not see why IPCC-AR4 is pointed out as a special landmark along the path of the development of global climate models or Earth system models. Rather the initial coupled model inter-comparison project (CMIP) would deserve to be mentioned, and the way the CMIPs were developed by the community and the working group on coupled modeling (WGCM).**

**Response**: We agree that CMIP and its promoter WGCM played an important role for climate modelling in the world. They should be more acknowledged than IPCC. The latter is however much more popular for a general public.

**L71: … Its performance is presented in a separated paper (Wu et al., to be submitted). … Please check if references to "to be submitted" articles are allowed in GMDD. Better write "… Its performance will be presented in a separate paper. ...".**

**Response**: Proposition relevant and accepted.

**L78: Please include a figure that compares the profiles of layer thickness against the height of the layer for the L26 and L46 grids, using for example a simple log-p height definition. This would clarify how the vertical resolution has been improved.**

**Response**: We added Figure 1 in the revised manuscript to show the profiles of layer thickness against the height for 26 vertical layers in BCC-CSM-1.1m and 46 vertical layers in BCC-CSM2-MR.

**L81: 2.1 Atmospheric component BCC-AGCM. This is the main documentation for the new version of the atmospheric model. Therefore I would expect to find here basic information for all processes. This should include the numerical techniques used in the dynamics, the transport scheme and the physics. This basic information should be kept concise, so that most of the room can still be devoted to the subsections (a) to (d) for the changes compared to the preceding documented version.**

**Response**: For the completeness of the documentation, we added more detailed descriptions on the model dynamics core, and all the models physics are summarized in Table 2.

**L96: … considered …→… considers …**
**L100: … environment, The mass …→… environment, the mass…**
**L128:… at each model grid …→ …in each model grid cell …?**
**L138: …T_con → T_conv …**

**Response:** Corresponding modifications are now included.

**L144: "… Following the method above, the cloud fraction (C_deep and C_shallow),temperature (T_deep and T_shallow), specific humidity (q_deep and q_shallow) for the deep convective, shallow convective clouds can be then deduced sequentially…." Does the scheme allow the concurrent occurrence of shallow and deep convection in the same atmospheric column, or is only one type of convection allowed at any one time in a single column? If concurrent occurrence is allowed, how is the parametrized computation of deep and shallow convection split, and how is the necessarily sequential diagnostics of C, T and q in the shallow and deep updrafts organized?**

**Response**: We have rewritten this part of the manuscript. In fact, shallow and deep convections can concurrently occur in the same atmospheric column at any time step. The shallow convection follows the deep convection, and it occurs at vertical layers where local instability still remains after the deep convection. In BCC models, the three moisture processes (i.e. deep convection, then shallow convection, and finally stratiform precipitation) is sequentially executed. That means, the model-box mean T and q are updated immediately after each process.

**L151 and L153: These equations can be numerically unstable in the limit of C_deep+C_shallow!1, because of the division by (1-Cdeep-Cshallow) that is needed to obtain the unknowns T_ambient and q_ambient. Is this a practical problem, or rendered irrelevant by the multiplication factor (1-C_conv) in Eq. 2?**

**Response:** We have added "if $C_{deep} + C_{shallow} > 1$, $C_{deep}$ and $C_{shallow}$ are then scaled to meet the condition $C_{deep} + C_{shallow} = 1.0$, and Cs=0." So we do not need to know the values of T_ambient and q_ambient to derive Cs using Eq. (10).

**L157: : : : RH_abmient : : : ! : : : RH_ambient : : :**
**L164: I cannot find the publication by 'Kristj ÌA̧ansson and Kristiansen [2000]'. Do you mean 'Kristjánsson et al., 2000'? Kristjánsson, J. E., J. M. Edwards, and D. L. Mitchell (2000), Impact of a new scheme for optical properties of ice crystals on climates of two GCMs, J. Geophys. Res., 105(D8), 10063–10079, doi:10.1029/2000JD900015.**
**L202: : : : k = 1.18 _ 106 cmˆ □1 secˆ □1 : : : ! : : : k = 1.18 _ 10ˆ6 cmˆ □1 secˆ □1 : : :**

**Response:** They are corrected now.

**L206: Section 'd. Parameterization of gravity wave drag' This paragraph discusses drag by dissipating gravity waves originating from flow over orography or atmospheric sources. What about drag by blocking effects from unresolved orography? Are such effects, which sometimes are included in gravity wave parameterizations, considered in BCC-AGCM3-MR?**

**Response:** The scheme of gravity wave drag generated from convective sources is that in Beres et al. (2004). The drag by blocking effects is still not involved in present version of BCC-AGCM3-MR.

**L221:... This parameterization scheme of convective gravity waves can improve the model's ability to simulate the stratospheric quasi-biennial oscillation in BCC-AGCM3-MR. … This is a rather general statement. It is clear that non-orographic gravity waves make a significant contribution to the forcing of the QBO, and if the gravity waves are not resolved, then their effect needs to be parameterized for the simulation of the QBO, and tunable parameters can be used to improve the structure of the QBO. Is the CF parameter, which you tune, valid for all latitudes or only for equatorial latitudes, where the QBO exists?**

**Response**: The convective fraction (CF) within a grid cell is an important parameter and is closely related to the deep convection process. It is tuned to obtain the right wave amplitudes. In BCC-AGCM3-MR, it is a constant and does not vary geographically. So it is valid for all latitudes where convection occurs.

**L277: Is the "simple scheme about the surface albedo, roughness length, turbulent sensible and latent heat fluxes over rice paddies" documented, or is there a manuscript in preparation? If not, and if the scheme is indeed simple, you should include the documentation here.**

**Response**: A new manuscript for this regard is indeed under preparation by our team in charge of land surface processes.

**L348: …The preindustrial climate state of BCC-CSM2-MR is preceded by a more than 500 years piControl simulation following the requirement of CMIP6. …Which were the goals of the spin-up simulation for the piControl experiment? Which were the criterions for declaring the spin-up phase completed? It would be interesting to learn about these criterions.**

**Response:** In the revised manuscript, we added some details about how the spinup was effectively accomplished. For this issue, we just followed the recommendations from CMIP project. Basically we check the steadiness of some globally-integrated quantities. We also added Fig.2 in the revised manuscript, showing energy balance at top of the atmosphere.and surface air temperature over the globe in the piControl simulation.

**L.353: : : : the up-limit of the atmosphere : : :! : : : the top of the model atmosphere : : :**
**Response**: Modified.

**L.362: …It means that the whole earth system in our models is very close to energy equilibrium. … For a transient period it is a bit difficult to judge from the similarity between TOA radiation fluxes of the model and the observations whether the model is**

generally in a "good" equilibrium. **How is the energy balance for the preindustrial control experiment? Here we know that the net energy flux at TOA should be zero except for fluctuations related to the internal variability of the coupled system. If the model system has energy leaks, as many climate models have, we should see this clearly in the stabilized piControl simulation. Such a leakage would have to be considered in the comparison of the transient TOA radiation fluxes of the model and the observations. My suggestion is that you discuss the net energy flux at TOA and the surface and maybe other quantities of interest of the piControl simulation before starting the discussion of the historical simulation. This could for instance be embedded in a new section that explains the tuning goals of the piControl experiment.**

**Response:** Yes, we agree with the referee's reasoning. It is a bit difficult to judge from the similarity between TOA radiation fluxes of the model and the observations. We have modified the description. In addition, following referee's suggestion, we have added Figure 2 (in the revised manuscript) in Section 3 to show the time series of annual mean of net energy flux at top of the atmosphere and the global sea surface temperature from 600 years piControl simulation. It means that the whole system in BCC-CSM2-MR nearly reaches its equilibrium after 600 years piControl simulation.

**L372: : : : These biases are reduced in BCC-CSM2-MR. : : : Which of the model changes discussed earlier cause the strong (and useful) changes in the tropical SW and LW cloud radiative forcing between the old and new model?**

**Response:** Modified. In low latitudes between 30 °S and 30 °N, BCC-CSM1.1m shows excessive cloud radiative forcing for both shortwave and longwave radiations. These biases are reduced in BCC-CSM2-MR and may be attributed to the new algorithm in diagnosing cloud fraction especially convective cloud amount.

**L379: : : : Biases of annual mean surface air temperature (at 2 meters) : : : Figure 2 shows the spatial patterns of the T2m bias. But first of all I am wondering how the transient global mean temperature is evolving from the stabilized pre-industrial mean temperature representative for 1850 to the present day. Please discuss first the global mean evolution before describing the pattern of the T2m bias near the end of the historical experiment. Further, it would be interesting to read your opinion on the contribution of the model changes to the observed differences between the models. Can you attribute the disappearance of the cold bias in the southern oceans in the new model compared to the old model to the changes in the model for turbulent fluxes over sea ice? Are specific changes of the land model important for the increased cold bias in east Asia and Siberia in the new model compared to the old model? For the land surface biases it would be valuable to know if these patterns are already present in AMIP simulations, where effects of oceanic biases are excluded.**

**Response:**
(1) Following referee's suggestion, we advanced two sections (old Section 4.4 and Section

4.5) to the position just after Section 4.1. They become now Section 4.2 and Section 4.3. The former presents the transient global mean temperature evolving from 1850 to the present day, and the latter (new Section 4.3) presents climate sensitivity to CO2 increasing.

(2) As shown in Figure 2 (in the initial manuscript, and renumbered to Fig.5 in the revised manuscript), the disappearance of the cold bias in the Southern Oceans in BCC-CSM2-MR (compared to BCC-CSM1.1m) is quite certainly attributable to the new scheme of cloud fraction parameterization implemented in BCC-CSM2-MR, which largely improves the low-level clouds simulation between 40 °S to 60 °S over the Southern Ocean (not shown).

(3) Biases over land surface in the two coupled models are similar to each other. They are furthermore already present in AMIP. So biases over land surface (at least partly) come from the land surface model.

**L386: annual mean precipitation What is the global mean precipitation in both models? Do you have any thoughts about the contribution of the changes in the deep convection scheme to the strong wet bias in the Maritime Continent?**

**Response**: The global mean precipitation rates in BCC-CSM1.1m and BCC-CSM2-MR are 2.87 mm/day and 2.94 mm/day, respectively. A precipitation rate of 2.68 mm/day is the 1986-2005 mean observed precipitation from Global Precipitation Climatology Project (Adler et al., 2003). The excessive rainfalls in the Maritime Continent seem amplified in BCC-CSM2-MR, which is attributed to abundant stratiform precipitation which accounts 39% of total precipitation in BCC-CSM2-MR. That percentage was 35% in BCC-CSM1.1m.

**L. 393: … The evaluation is done against climatology of ERA-Interim … NCEP dataset … From the text and the figure caption it seems rather that only NCEP is used and not ERA-Interim. Can you please clarify this?**

**Response**:It was a mistake. In fact, ERA-Interim is used as a reference state instead of NCEP.

**L421:…Given a much higher vertical resolution and an advanced parameterization of the gravity wave drag … Despite the more complete parameterization of gravity wave drag – now including gravity waves from atmospheric sources – the zonal mean zonal wind biases in the high latitudes of the stratosphere have increased near 10 hPa, where one would expect the main benefit from gravity wave parameterizations. Can you explain why there is no benefit from the improved GWD parameterization on the structure of the polar night jets? Did you attempt to tune the gravity wave drag to reduce errors in the stratospheric extratropical zonal circulation?**

**Response:** We don't know the exact cause for the zonal wind biases in the stratosphere of high latitudes in BCC-CSM2-MR. Maybe the lack of gravity wave drag generated by atmospheric blocking is the explanation. We expect to reduce the bias in next version by adding this process.

**L427: … In Figure 6(b), the BCC-CSM2-MR simulations present a clear quasi-biennial oscillation of the zonal winds as observed. …The downward propagation … does not penetrate to sufficiently low altitudes. … Though the vertical resolution is increased, it is still too low to expect a QBO simulation down to at least 70 hPa, because the forcing from resolved waves cannot be adequately resolved. Therefore it seems like the QBO occurring in the new model must be dependent entirely or nearly entirely on the Beres parameterization. Has this scheme been tuned to obtain the QBO in the new model?**

**Response:** Yes, we agree with the referee's arguments. The number of layers (46) in our model is certainly too low to resolve adequately vertically-propagated waves. But we have not tuned the Beres scheme that seems to work well in our new model. This issue deserves further studies in the future.

**L470: : : : The most remarkable improvements of BCC-CSM2-MR appear in the boreal warm seasons, : : : To which model improvement do you attribute the strong improvement of SIE or SIC?**

**Response:** It is hard to say. Those improvements may be partly contributed by many aspects of new model physics schemes such as turbulent flux over sea ice and ocean surfaces, cloud fraction, or atmospheric circulation improvement at high latitudes.

**L495: : : : Our CMIP6 model can capture this warming hiatus. : : : The word "capture" suggests that the hiatus is a predictable climate feature that a coupled climate simulation can be expected to reproduce if the forcing is realistic and the model is "correct". Is this what you want to express? Maybe it is better to write for instance: "The historical simulation of the CMIP6 model shows a hiatus towards the end of the simulation that resembles the observed one." Do you have other ensemble members for the historical CMIP6 simulation, and if so do all members reproduce the hiatus of 1998-2013? The figure shows also that the CMIP5 simulation is significantly colder in the early decades than observations or the CMIP6 simulation. Later on, however, both simulations evolve by and large in a similar way. Can this be explained by the external forcing (volcanic aerosols?)? Do you have any insight that you can share?**

**Response:** To address this comment, we need to point out three elements. (1) The phrase concerning the simulated warming hiatus has been changed in the revised manuscript as suggested. (2) As many CMIP diagnostic papers, we also used only one member in our manuscript. That member shows hiatus. Recently, two additional members of the historical simulation are also available. As shown in new Figure 4, two members show hiatus towards the end of the simulation. (3) Both BCC models seem too sensitive to volcanic aerosols. But we do not fully understand the behaviors of global mean temperature curves.

**L509: …Observation-based NSIDC data are also plotted when available. … The caption for Fig. 11 reads: ": : : observations-based Hadley Centre Sea Ice and Sea Surface Temperature data set (Rayner et al., 2003)." Please clarify.**

**Response:** Modified. In Figure 11 (in the initial manuscript, and renumbered to Fig.14 in the revised manuscript), the observed sea-ice extent are derived from Hadley Centre Sea Ice and Sea Surface Temperature data set. In Figure 8 (now renumbered to Fig.15), the observed seasonal cycles of sea-ice extent in (a) and (b) are based on the National Snow and Ice Data Center (NSIDC; Fetterer et al., 2002) data sets. In order to keep consistency, the NSIDC data in Figure 8 (now renumbered to Fig.15) is replaced by Hadley data.

**L516: 4.5 Climate sensitivity to CO2 increasing Figure 12 following Gregory (2012) not only provides estimates for the ECS, but primarily provides information on the climate feedback. Comparing both models, the ECS is similar, but the feedback parameter is substantially different: BCC-CSM3-MR: ca. -1 W/m2/K; BCC-CSM1.1m: ca. -1.3 W/m2/K Thus the result that both ECS values are very similar results only because the initial 4xCO2 forcing is quite different: BCC-CSM3-MR: ca. 6 W/m2; BCC-CSM1.1m: ca. 7.5 W/m2/K Can you please comment on the origin of the large difference in the initial forcing?**

Response: Yes, that's an interesting point. We added a paragraph in the revised manuscript concerning the 4xCO2 initial forcing, feedbacks and ECS. Due to changes of atmospheric profiles (temperature, water vapor and cloud), it is possible to have different forcing for a quadruple CO2, and even the radiative transfer is unchanged. Feedbacks operate certainly in different ways in the two models. What is interesting is that the final ECS converges between the two models.

**L529: : : : abruptCO2 : : : ! : : : abrupt4xCO2 : : : ?**

Response: Modified.

---

## Author Comment (AC2) · 10 Feb 2019

Dear Referee #2,

We would like to thank you for your comments and suggestions to improve the quality of our manuscript "The Beijing Climate Center Climate System Model (BCC-CSM): Main Progress from CMIP5 to CMIP6" by Tongwen Wu et al. Following your suggestions, we added 5 further figures in the revised manuscript. So, all figures in the first manuscript are renumbered. We have also changed the order of presentation for some subsections in Section 4 (Results). The point-to-point responses to your comments are enclosed in the following.

Best regards,

Tongwen Wu and all co-authors

========================================
**Response to Anonymous Referee #2**

**This is a description paper of BCC-CSM2-MR. The upgraded schemes are well described, while evaluations for internal variability and long-term trends are insufficient. Authors emphasized improvements in several aspects. Some of them (e.g., representation of QBO) are remarkable improvements but some others are not very convincing. Major revisions and some additional analyses are required as follows:**

**Major comments:**
**The paper lacks the information of ensemble size of each historical runs. The following two comments are related to this issue:**

**1.1 Is the global warming trend of CMIP6 model shown in Fig. 10 the ensemble-mean value? What is the gray shading? Maximums/minimums of CMIP5? If the ensemble size of CMIP6 is only one member, you cannot deny the possibility that the simulated slowdown after 1998 happened by chance due to natural variability. On the other hand, if it is ensemble-mean value, then the slowdown after 1998 implies it is the result of external forcing. In the latter case, you need to discuss whether the difference in the trends between CMIP5 and CMIP6 is due to the difference in external forcing or the model update.**

**Response:**
The mentioned curves (in old Fig. 10 or new Fig. 4) are from a single realization for both BCC models (CMIP5 and CMIP6). After the initial submission of the manuscript, two other members are now available for our CMIP6 exercises, which are added in the new Fig. 4. Gray shaded area shows the spread calculated from 31 CMIP5 models. The caption in Fig. 10 of the initial manuscript was not clear, and is now modified in the new manuscript. Concerning the global warming hiatus or slowdown after 1998 in old Figure 10 (or new Fig.4, it is the overlay

of internal variability and long-term trends. We need to explore it further in the future. Among the three members, two show some features of hiatus.

**1.2 Is the improvement of MJO shown in Fig. 7 robust? A correlation coefficient depends on a sample size. Are the sample sizes (the total numbers of years) you used for the analyses of the observation, BCC-CLM1.1m, and BCC-CLM2-MR comparable? If the improvement is true, what do you think is a factor improving the representation of MJO? New convection scheme?**

**Response:** Yes, MJO activities largely depend on period and time length of analysis. But we believe that our analysis in Figure 7 is robust and consistent: all data for observation, BCC-CSM1.1m, and BCC-CSM2-MR are from the same period from 1997 to 2005. We believe that the improvement of MJO simulation in BCC-CSM2-MR is true and can be possibly attributed to deep convection scheme (Table 2). Further investigations are underway.

**2 It seems that the results in Fig. 8 and Fig. 11 are inconsistent. Is it due to the difference between the reference datasets (NSIDC, ECMWF forecasts, and Hadley Centre Sea Ice)? Please add some explanation to figure out it and the reason why you used those datasets.**

**Response:**
Figure 11a and 11b (in the old version of manuscript) shows seasonal cycle of sea-ice extent climatology, in which the observation data is derived by National Snow and Ice Data Center (NSIDC; Fetterer et al., 2002) and are directly downloaded from
https://svn-ccsm-models.cgd.ucar.edu/tools/proc_ice/trunk/ice_diag/data/SSMI.ice_extent.1981-2005.monthly.regional.txt. Figure 11c and 11d (in the old version of manuscript) shows seasonal cycle of sea-ice thickness averaged for the Northern Hemisphere and the Southern Hemisphere, which is computed based on 1980-2005 global monthly $1/4 °\times 1/4 °$ gridded dataset based on European Center for Medium-Range Weather Forecast (Tietsche, et al., 2014).

Figure 8 (in the initial manuscript) presents long term change of sea-ice extent from 1851 to 2012, the observed data are computed based on global monthly $1 °\times 1 °$ gridded dataset of Hadley Centre Sea Ice and Sea Surface Temperature (Rayner et al., 2003).

The observed sea-ice extent in Figs. 11 and 8 (in the initial manuscript) doesn't come from a same source. NSIDC data are mainly derived from satellite observations, so with a higher quality. But they cover a too-short period. This explains why Fig. 8 in the initial manuscript the long-lasting Hadley Centre Sea Ice and Sea Surface Temperature data set. For the sake of consistency, we decided to use only Hadley center data throughout the manuscript.

**3 Section 4.2: Please discuss the SST cold bias in the equatorial Pacific and the double ITCZ problem. Evaluations for the subsurface ocean (temperature, zonal currents) are also required.**

**Response:**

We added two more figures to discuss the SST cold bias and the interannual variations of NINO3.4 SST in the equatorial Pacific. As length of the paper is limited and there are already 20 figures, the subsurface ocean (temperature, zonal currents) will be explored in the future.

**4 Section 4.6: There is a large seasonal cycle in the East Asian climate. Analyses should be done for each season, especially for JJA and DJF.**

**Response:** We added a further figure to discuss the DJF and JJA precipitation in East Asia.

**5 Comparison of ENSO representation (NINO3.4 time series, amplitude, spatial pattern etc.) is necessary to evaluate the model performance.**

**Response**: We added a figure to show behaviors of Nino3.4 time series. The detail evaluation for ENSO representation will be analyzed in other papers.

**Minor issues:**

**6 L19: models -> model's**

**7 L27-28 "Compared to BCC CMIP5 models, BCC CMIP6 models show: : :": The expression "models" is inappropriate. This paper compared only one model for each (BCC-CSM1.1m for CMIP5 and BCC-CSM2-MR for CMIP6).**

**8 L41: More -> more**

**9 L53: The full name of CMIP5 and Taylor et al. (2012) should appear at L42.**

**10 L59: Please comment on Section 5 and 6.**

**11 L64: Please add the full name for NCAR.**

**12 L64: Coupler -> coupler**

**Response:** 6-12, all done.

**13 L65-66: Are tuning parameters also the exactly same between the two models?**

**Response:** Some physical schemes in the two models are not the same. Tuning parameters are not exactly the same, neither.

**14 L79: Please add the level of the top layer.**

**Response:** Added as suggested. The tops of atmosphere in BCC-CSM1.1m and BCC-CSM2-MR are at 2.917hPa and 1.459 hPa, repsecctively.

**15 L240: What is the CEVSA model?**

**Response:**CEVSA model is the carbon exchange between vegetation, soil and the atmosphere (CEVSA) model

**16 L243: Table 2 -> Table 3**

**Response:** Modified.

**17 L362: 2014 -> 2005? The end of the CMIP5 historical run is 2005.**

**Response:** Modified. In Table 4, and Fig.1 (renumbered to Fig.3 in the revised manuscript), model results are for the period 1986–2005, while the available CERES-EBAF data are for

2003–2014.

**18 L371 "1986-": "1985-" in the caption of Fig. 1**
**Response:** Modified.

**19 L374-376: In the mid latitude, discrepancies in the two observations are also large. You cannot discuss the difference between the two models.**
**Response:** Modified as suggested.

**20 L396: Precipitation data diagnosed in reanalysis is not necessarily correct. You had better use the observation-based dataset.**
**Response:** Modified as suggested.

**21 L406: Figure 4s -> Figure 4**
**Response:** Modified.

**22 L409: Were there any reason you used the two different reanalysis datasets (NCEP and ERA-interim) for each analysis?**
**Response:** Modified. All the reanalysis data used to draw figures are now replaced by ERA-interm.

**23 L437 "forcing is less adequate": What forcing?**
**Response:** "forcing" denotes gravity wave forcing. Modified to "The amplitudes of the QBO cycles in the simulation are weaker than in the reanalysis, which is possibly due to inadequate gravity wave forcing to drive the QBO."

**24 L452: improvementcompared -> improvement compared**
**Response:** Modified.

**25 L478-481: Is the lower and deeper NADW better? Please show the observation based values.**
**Response:** Modified. The observation-based value is 25 Sv in Talley et al. (2013).

**26 L486 "from 60S to 60N": Why did not you use the average from 90S to 90N?**
**Response:** Only the area from 60°S to 60°N is averaged in Fig. 10 (renumbered to Fig.4 in the revised manuscript). This is mainly motivated by the consideration that HadCRUT4 dataset had very few observations in polar regions in earlier 20th century.

**27 L491, L500: HadCRU -> HadCRUT4**
**Response:** Modified.

**28 L503: Figure 2c -> Figure 10?**
**Response:** Modified.

**29 L529: abruptCO2 -> abrupt CO2**

**Response:** Modified.

**30 L535 "the TCR of the new version model BCC-CSM2-MR is lower than BCCCSM1.1m,": lower -> higher? It is inconsistent with L527-528.**

**Response:** The TCR of the new version model BCC-CSM2-MR is lower than BCC-CSM1.1m, but the ECS of BCC-CSM2-MR is slightly higher than BCC-CSM1.1m.

**31 L566-567: Are there any impact of the upgraded land surface scheme on the improvement of the rainfall diurnal cycle?**

**Response:** We didn't explore this issue, but we think the upgraded land surface scheme has minor role for the improvement of the rainfall diurnal cycle.

**32 L616-620: Please see 31.**

**Response:** Modified.

**33 Table 3 "(Guenther et al., 2012)": Please check the font.**

**Response:** Modified.

**34 Table 4: What is the value of the net energy at TOA for OBS? 0.81?**

**Response:** The net energy at TOA is 0.81 W $m^{-2}$ for CERES-EBAF and 5.73 W $m^{-2}$ for CERES data.

**35 Table 4, Notes, "1981-2014": 1981-2005 for BCC-CSM1.1m? Why are the periods slightly different among the figures or tables (1985-2005 for Fig. 1, 1986-2005 for Fig. 2-3 and 5, 1980-2005 for Fig. 4, 8-9, 13)?**

**Response:** Yes, that's done. We apologize for the inconsistency. However, In Table 4 and Fig. 1 (renumbered to Fig.3 in the revised manuscript), model data are the mean from 1986 to 2005, while observation is only available for 2003–2014.

**36 Figure 7 d-f: Please show the range of longitude used for the average to obtain the latitude-time section.**

**Response:** Modified.

**37 Figure 14: The result of BCC-CSM1.1m (BCC-CSM2-MR) should be in the bottom (middle) as same as in the other figures.**

**Response:** Yes, that's done. We appologize for the inconsistency.

---

## Referee Report (RR1)

Review of the revised version of 'The Beijing Climate Center Climate System Model (BCC-CSM): Main Progress from CMIP5 to CMIP6' by Wu et al.

General comments

The authors have addressed the issues from the review of the 1st version of their manuscript. In the following there are a number of comments and requests for clarification related to modified or new parts of the manuscript. Once these points are clarified and the manuscript is revised accordingly, I recommend its publication, so that this model documentation paper becomes available to the community.

Detailed comments

L250... : "... In BCC-AGCM3-MR, the gravity wave drag generated from convective sources is introduced as in Beres et al. (2004), but drag by blocking effects is still not involved. ... "
The Beres scheme parameterizes convective sources for the parameterized gravity wave drag. What about other sources, which may be important for the higher latitudes?
Has the McFarlane scheme been tuned for the model simulations?

L420: "... Figure 2 ... The whole system in BCC-CSM2-MR nearly reaches its equilibrium after 600 years. ... "
Thanks for adding Figure 2. You claim that the system has nearly reached an equilibrium after 600 years. But from Figure 2 one cannot distinguish, whether the system is still equilibrating, or the system has already equilibrated in combination with an energy leak/source of ca. +0.4 $W/m^2$, or whether the equilibration is still ongoing in presence of a leak/source. What can be concluded from this figure is that the net TOA energy flux fluctuates without obvious trend around ca. 0.4 $W/m^2$, while the SST of the 2nd 300 year period is a bit higher than in the 1st one.
Additionally, I would like to ask the authors to display the global mean surface (or near surface) temperature, either in place of the SST or in addition. The main reason is that the Gregory plots (as in your Figure 6), which are relevant for judging "equilibration", relate the radiative forcing at TOA to the global mean surface temperature, not to SST.

L458-459: "... When a 9-year smoothing is applied ... "
The caption of Figure 4 writes: "... of 11-year smoothed ... "
Please clarify.

L466-467: "... There are two members (r1i1p1f1 and r2i1p1f1 in Fig. 4) of historical simulations of the CMIP6 model show a hiatus towards the end of the simulation that resembles the observed one ... "
r3i1p1f1 is not comparable to the observed hiatus. This realization has a short spell of colder years centered at 2010, which is different to the lack of warming in the observational record and in r1i1p1f1 from ca. 2000 to the end of the simulations.
Please modify your text.

L472... : As you describe the earlier hiatus appears in r2, but not in r1, and the later hiatus occurs in r1, but not in r2. This clearly excludes any simple response to forcing, and makes internal variability a much more likely reason.

L485... : Figure 5 shows in most places between 60S and 60N a warm bias, so that the area average likely is also positive, i.e. the model is rather warmer than ERA-Interim. In Figure 4, however, the

60S to 60N average of the r1 simulation almost always is below the observed record, especially due to the strong response to the Mt. Pinatubo eruption. How does this fit together?

L528-529: Is the similarity of the ECS by chance, or was this a goal of the tuning procedure? Please clarify this in the manuscript.

L576… : "… The zonal mean of zonal wind biases in the high latitudes of the stratosphere in BCC-CSM2-MR have increased near 10 hPa, where model biases may be partly caused by not yet involved gravity wave drag that generated by blocking effects. … "
Blocking mainly affects the tropospheric circulation in regions of steep topography. The polar night jets are rather influenced by gravity wave drag. In any case, the too strong polar night jets indicate a too weak drag. Maybe some non-orographic gravity wave sources are not represented, see earlier comment.

L596: "… which is possibly due to inadequate gravity wave forcing to drive the QBO. … "
This is one possibility. But there is also the other one: The wave-meanflow interaction based on resolved waves (Kelvin waves, mixed Rossby-gravity waves, … ) is probably not realistic. One reason that would contribute to such a deficiency is the relatively coarse vertical resolution that would affect the vertical wave lengths and the wave damping.
(For the planned QBO-article: The separate analysis of resolved wave meanflow interaction and parameterized wind tendencies in the QBO domain, together with advective tendencies, will be important to understand the nature of the simulated QBO.)

L633: "… the period ENSO periodicity … " → "… the mean period of ENSO … "

---

## Author Response (AR2)

Dear Dr. Juan Antonio Añel,

We hereby submit, for your consideration, the second revision of manuscript entitled "The Beijing Climate Center Climate System Model (BCC-CSM): Main Progress from CMIP5 to CMIP6". We have incorporated all comments from two Referees in the revised manuscript. The point-to-point responses to Referees are enclosed in the following.

Thanks for considering this paper, and we look forward to hearing from you regarding its disposition.

Best regards,

Tongwen Wu, and all Co-Authors

We thank both reviewers for their constructive and insight comments. We have carefully addressed these comments in the revised manuscript. All our responses are in Italic Font in the following point-to-point reply.

Response to Anonymous Referee #1

**General comments**

The authors have addressed the issues from the review of the 1st version of their manuscript. In the following there are a number of comments and requests for clarification related to modified or new parts of the manuscript. Once these points are clarified and the manuscript is revised accordingly, I recommend its publication, so that this model documentation paper becomes available to the community.

**Detailed comments**

L250... : "... In BCC-AGCM3-MR, the gravity wave drag generated from convective sources is introduced as in Beres et al. (2004), but drag by blocking effects is still not involved. ..." The Beres scheme parameterizes convective sources for the parameterized gravity wave drag. What about other sources, which may be important for the higher latitudes? Has the McFarlane scheme been tuned for the model simulations?

**Response:** We have rewritten this paragraph and added more descriptions "Gravity waves generated by topography and fronts are important for higher latitudes. The efficiency parameter in the McFarlane scheme is set to 0.125 in BCC-AGCM2.2 and doubled to 0.25 in BCC-AGCM3-MR to obtain a better result of the polar night jet. In future, it is planned to improve the orographic gravity wave scheme and to implement parameterizations of gravity waves emitted by fronts and jets."

L420: "... Figure 2 ... The whole system in BCC-CSM2-MR nearly reaches its equilibrium after 600 years. ... "Thanks for adding Figure 2. You claim that the system has nearly reached an equilibrium after 600 years. But from Figure 2 one cannot distinguish, whether the system is still equilibrating, or the system has already equilibrated in combination with an energy leak/source of ca. +0.4W/m2, or whether the equilibration is still ongoing in presence of a leak/source. What can be concluded from this figure is that the net TOA energy flux fluctuates without obvious trend around ca. 0.4 W/m2, while the SST of the 2nd 300 year period is a bit higher than in the 1st one. Additionally, I would like to ask the authors to display the global mean surface (or near surface) temperature, either in place of the SST or in addition. The main reason is that the Gregory plots (as in your Figure 6), which are relevant for judging "equilibration", relate the radiative forcing at TOA to the global mean surface temperature, not to SST.

**Response:** We followed the suggestion and replaced SST by the global surface air temperature in Figure 2b. The corresponding paragraph was rephrased.

L458-459: "... When a 9-year smoothing is applied ... " The caption of Figure 4 writes: "... of 11-year smoothed ... " Please clarify.

*Response: Thanks. We checked our script and the caption in Figure 4 is right. The filter uses 11 years.*

L466-467: "... There are two members (r1i1p1f1 and r2i1p1f1 in Fig. 4) of historical simulations of the CMIP6 model show a hiatus towards the end of the simulation that resembles the observed one  $\cdots$ " r3i1p1f1 is not comparable to the observed hiatus. This realization has a short spell of colder years centered at 2010, which is different to the lack of warming in the observational record and in r1i1p1f1 from ca. 2000 to the end of the simulations. Please modify your text.

Response: We have reworded this sentence to make our point clearer. The revised text is "Although the third member (r3i1p1f1) simulated a global warming slowdown from 2004 to 2012, it is not comparable to the observed hiatus as it has a short spell of colder years centered at 2010.".

L472... : As you describe the earlier hiatus appears in r2, but not in r1, and the later hiatus occurs in r1, but not in r2. This clearly excludes any simple response to forcing, and makes internal variability a much more likely reason.

Response: Thanks for your comments. We have rewritten this sentence as "So, the simulation of global warming hiatus in BCC CMIP6 model clearly excludes any simple response to forcing, and makes internal variability a much more likely reason".

L485... : Figure 5 shows in most places between 60S and 60N a warm bias, so that the area average likely is also positive, i.e. the model is rather warmer than ERA-Interim. In Figure 4, however, the 60S to 60N average of the r1 simulation almost always is below the observed record, especially due to the strong response to the Mt. Pinatubo eruption. How does this fit together?

Response: This is only a problem of reference. Figure 5 shows the surface air temperature as a difference to ERA-Interim during the same period of 1986 to 2005. But Figure 4 shows the anomalies of each dataset relative to its own climatology of 1961-1990.

L528-529: Is the similarity of the ECS by chance, or was this a goal of the tuning procedure? Please clarify this in the manuscript.

**Response: This is entirely fortuitous. We added a phrase in the revised manuscript "We remind that this is a pure coincidence since we did not intentionally tune our model for its sensitivity".**

L576... : "... The zonal mean of zonal wind biases in the high latitudes of the stratosphere in BCCCSM2-MR have increased near 10 hPa, where model biases may be partly caused by not yet involved gravity wave drag that generated by blocking effects. ... " Blocking mainly affects the tropospheric circulation in regions of steep topography. The polar night jets are rather influenced by gravity wave drag. In any case, the too strong polar night jets indicate a too weak drag. Maybe some non-orographic gravity wave sources are not represented, see earlier comment.

Response: We agree with this argument and have rewritten the corresponding paragraph as "The too-strong polar night jets clearly indicate an insufficient atmospheric drag at this level. It may be partly caused by the lack of effects in relation to some non-orographic gravity waves generated by atmospheric fronts and jets. We expect to reduce this model bias in next version by adding this process."

L596: "... which is possibly due to inadequate gravity wave forcing to drive the QBO. ... " This is one possibility. But there is also the other one: The wave-meanflow interaction based on resolved waves (Kelvin waves, mixed Rossby-gravity waves, ... ) is probably not realistic. One reason that would contribute to such a deficiency is the relatively coarse vertical resolution that would affect the vertical wave lengths and the wave damping. (For the planned QBO-article: The separate analysis of resolved wave meanflow interaction and parameterized wind tendencies in the QBO domain, together with advective tendencies, will be important to understand the nature of the simulated QBO.)

Response: We agree with your argument, and we hope that we will be able to improve QBO simulations by this approach.

L633: "•••• the period ENSO periodicity ••• "  $\rightarrow$  "•••• the mean period of ENSO ••• "

Response: Modified.

**Response to Anonymous Referee #2**

I am almost satisfied with the authors' revision. They have honestly responded to my comments. I have only minor comments as follows:

1. L74-75 "They differ only by their horizontal resolutions": The versions of the ocean and sea ice models are also different according to Table 1.

Response: Thank you, the "only" should be replaced by "mainly".

2. L544 and L608: Omit the bullet. *Response: Modified.*

3. L584 "wind velocity at 850 hPa (U850 and V850) and at 200 hPa (U200)" >> "zonal and meridional wind velocity at 850 hPa (U850 and V850) and zonal wind velocity at 200hPa (U200)"

Response: Modified.

- 4. L587 "precipitation and zonal wind at 850 hPa" >> "PRCP and U850" *Response: Modified.*
- 5. L589 "geopotential height at 500 hPa" >> "Z500" *Response: Modified.*
- 6. L592 "precipitation" >> "PRCP" *Response: Modified.*

7. L658-659 "cold SST in the eastern equatorial Pacific still extends too far west in both models and a cold tongue bias exists in the equatorial Pacific": It appears that the cold tongue cold bias is worse in BCC-CSM2-MR. Please touch on the issue.

Response: Yes, this is a deterioration. We modified the revised manuscript as "However, cold SST in the eastern equatorial Pacific still extends too far west in both models and a cold tongue bias exists in the equatorial Pacific and even gets a little worse in BCC-CSM2-MR. The annual mean SST in the coldest center near 110 W in the equatorial Pacific is below 23 °C in BCC-CSM2-MR, a deterioration compared to BCC-CSM1.1m.".

8. L670-671 "underestimated air-sea coupling strength": This is inconsistent with the fact that ENSO amplitude is overestimated in the models compared to the observation.

**Response: Please see next response.**

9. L672-673 "zonal advective feedback and thermocline feedback": It is known that the longer (4~5 year) ENSO period is related to the thermocline feedback rather than the zonal advection feedback (Guilyardi 2006).

Guilyardi, E., 2006: El Nino–mean state–seasonal cycle interactions in a multi-model ensemble. Climate Dyn., 26, 329–348.

**Response: In terms of potential causes for the too-short periodicity of the ENSO cycle, we took the reference to the results of Lu and Ren (2016) which might be inconsistent with other researches. In order to avoid confusion, we deleted the phrases in question in the revised manuscript.**

10. Figure 13: The characteristic of ENSO irregularity is improved in BCC-CSM2-MR in comparison to BCC-CSM1.1m.

**Response: We have added "The characteristic of ENSO irregularity is improved in BCC-CSM2-MR in comparison to BCC-CSM1.1m".**

11. Plots of sea-ice extent in Fig. 14 and 15 are still inconsistent each other. In Fig. 14a (the Arctic), BCC-CSM1.1m underestimates sea-ice extent compared to the Hadley data, although it overestimates in Fig. 15a (March). In the Antarctic, sea-ice extent simulated by BCC-CSM2-MR is too small in Fig. 14b, although the bias in Fig. 15b (March) is not so serious.

**Response: Thanks for your careful evaluation and checking. There were indeed inconsistences in old Figs. 14 and 15, entirely due to our confusion in plotting figures and manipulating different data sources. In the revised manuscript, we have updated them, and also modified the corresponding text.**

12. L784-785 "The center of precipitation around Japan is also well simulated in BCC-CSM2-MR": Distribution of precipitation around Japan in Fig. 18 appears to be strange. The rain-band should develop during summer, while the simulated rain-bands active in winter.

Response: This was partly induced by an inconsistency of color bars in Fig. 18 (summer versus winter). In order to avoid confusion, we replotted it using a same color bar.

**The Beijing Climate Center Climate System Model (BCC-CSM): Main Progress from CMIP5 to CMIP6**

3

4 Tongwen Wu1\*, Yixiong Lu1, Yongjie Fang1, Xiaoge Xin1, Laurent Li1,2, Weiping Li1, Weihua Jie1,

5 Jie Zhang1, Yiming Liu1, Li Zhang1, Fang Zhang1, Yanwu Zhang1, Fanghua Wu1, Jianglong Li1,

6 Min Chu1, Zaizhi Wang1, Xueli Shi1, Xiangwen Liu1, Min Wei3, Anning Huang4, Yaocun Zhang4,

7 Xiaohong Liu1,5

8

9 1Beijing Climate Center, China Meteorological Administration, Beijing, China

[revised manuscript text omitted]

(4)

$$C_{conv}T_c + (1 - C_{conv})\overline{T} = \overline{T}_{conv}$$
(5)

163 and

159

162

164

146

$$q^{*}(T_{c}) = q^{*}(\overline{T}) + \frac{\partial q^{*}(\overline{T})}{\partial \overline{T}}(T_{c} - \overline{T})$$

$$\tag{6}$$

where  $\bar{q}$  and  $\bar{T}$ ,  $\bar{q}_{conv}$  and  $\bar{T}_{conv}$  denote the model grid box-averaged water vapor mixing ratio and temperature in the 'environment' before and after convection activity, respectively.  $T_c$  and  $q^*(T_c)$  are the temperature inside the convective cloud plume and its saturated water vapor mixing ratio. Here, we assume that the shallow and deep convection can concurrently occur in the same atmospheric column at any time step. That is, the shallow convection scheme follows the deep convection and occurs at vertical layers where local instability still remains after deep convection.

171 If no supersaturation exists in clouds, we can obtain from Eqs. (4) and (5)

172
$$C_{conv} = \frac{\left(\overline{q}_{conv} - \overline{q}\right) - \frac{\partial q^{*}(T)}{\partial \overline{T}} \left(\overline{T}_{conv} - \overline{T}\right)}{q^{*}(\overline{T}) - \overline{q}}.$$
 (7)

[revised manuscript text omitted]

397
$$z_0(mm) = \begin{cases} 0.1 & \text{for } T_s \le -2^{\circ}C \\ 0.8 & \text{for } T_s > -2^{\circ}C \end{cases},$$
(19)

where  $T_s$  represents surface temperature. For the scalar roughness lengths, a theoretical-based model proposed by Andreas (1987) is used in the new scheme. This model expresses the scalar roughness  $z_s$  400  $(z_T \text{ or } z_0)$  as a function of the roughness Reynolds number R\*, i.e.,

401
$$\ln(z_s/z_0) = b_0 + b_1(\ln R_*) + b_2(\ln R_*)^2.$$
(20)

402 Andreas (1987, 2002) tabulates the polynomial coefficients  $b_0$ ,  $b_1$  and  $b_2$ .

403 3. Experimental design

All BCC simulations presented in this work follow the protocols defined by CMIP5 and CMIP6. We pay attention for them to be comparable in spite of showing the transition of our climate system model from CMIP5 to CMIP6. The principal simulation to be analyzed is the historical simulation (hereafter historical) with prescribed forcings from 1850 to 2005 for CMIP5 (to 2014 for CMIP6).

Historical forcings data are based as far as possible on observations and downloaded from the 408 webpage (https://esgf-node.llnl.gov/search/input4mips/). They mainly include: (1) GHG concentrations 409 (only CO2, N2O, Ch4, CFC11, CFC12 used in BCC models) with zonal-mean values and updated 410 monthly; (2) Yearly global gridded land-use forcing; (3) Solar forcing; (4) Stratospheric aerosols (from 411 volcanoes); (5) CMIP6-recommended anthropogenic aerosol optical properties which is formulated in 412 terms of nine spatial plumes associated with different major anthropogenic source regions (Stevens et 413 al., 2017). (6) Time-varying gridded ozone concentrations. In addition, aerosol masses based on CMIP5 414 415 (Taylor et al., 2012) are used for on-line calculation of cloud droplet effective radius in BCC model.

The preindustrial initial state of BCC-CSM2-MR is preceded by a 500-years piControl simulation 416 following the requirement of CMIP6. The initial state of the piControl simulation itself is obtained 417 through individual spin-up runs of each component of BCC-CSM2-MR in order for piControl 418 419 simulation to run stably and fast to its model equilibrium. Actually, the initial states of atmosphere and 420 land are obtained from a 10-years AMIP run forced with monthly climatology of sea surface temperature (SST) and sea ice concentration, and the initial states of ocean and sea ice are derived from 421 a 1000-years forced run with a repeating annual cycle of monthly climatology of atmospheric state from 422 the Coordinated Ocean-Ice Reference Experiment (CORE) dataset version 2 (Danabasoglu et al., 2014). 423 424 Figure 2 shows time series of the annual and global mean of net energy flux at top of the atmosphere (TOA) and the sea surface temperature for 600 years in the piControl simulation. The whole system in 425 BCC-CSM2-MR nearly-fluctuates around +0.4 W m-2 net energy flux at TOA without obvious trend 426 reaches its equilibrium after in 600 years. The global mean surface air temperature hashows a little 427

| 428 | bitsmall warming after 600 years (Fig. 2b). During the last $arc = 1000$ years, there $arc = 1000$ s $arc = 10$ |
|-----|--------------------------------------------------------------------------------------------------------------------------------------------------------------------------------------------------------------------------------------------------------------------------------------------------------------------------------------------------------------------------------------------------------------------------------------------------------------------------------------------------------------------------------------------------------------------------------------------------------------------------------------------------------------------------------------------------------------------------------------------------------------------------------------------------------------------------------------------------------------------------------------------------------------------------------------------------------------------------------------------------------------------------------------------------------------------------------------------------------------------------------------------------------------------------------------------------------------------------------------------------------------------------------------------------------------------------------------------------------------------------------------------------------------------------------------------------------------------------------------------------------------------------------------------------------------------------------------------------------------------------------------------------------------------------------------------------------------------------------------------------------------------------------------------------------------------------------------------------------------------------------------------------------------------------------------------------------------------------------------------------------------------------------------------------------------------------------------------------------------------------------|
| 429 | amplitude) oscillations of centennial scale averaged for the whole globe and a little weak for the areal                                                                                                                                                                                                                                                                                                                                                                                                                                                                                                                                                                                                                                                                                                                                                                                                                                                                                                                                                                                                                                                                                                                                                                                                                                                                                                                                                                                                                                                                                                                                                                                                                                                                                                                                                                                                                                                                                                                                                                                                                       |
| 430 | average of 60 S to 60 N in centennial scale. It is, to a large extent, They are certainly caused by internal                                                                                                                                                                                                                                                                                                                                                                                                                                                                                                                                                                                                                                                                                                                                                                                                                                                                                                                                                                                                                                                                                                                                                                                                                                                                                                                                                                                                                                                                                                                                                                                                                                                                                                                                                                                                                                                                                                                                                                                                                   |
| 431 | variation inof the whole system.                                                                                                                                                                                                                                                                                                                                                                                                                                                                                                                                                                                                                                                                                                                                                                                                                                                                                                                                                                                                                                                                                                                                                                                                                                                                                                                                                                                                                                                                                                                                                                                                                                                                                                                                                                                                                                                                                                                                                                                                                                                                                               |

**432 4. Evaluation and comparison between BCC CMIP5 and CMIP6 models**

**433 **4.1 Global Energy Budget**

[revised manuscript text omitted]